# Revisiting and attributing the global controls on terrestrial ecosystem functions of climate and plant traits at FLUXNET sites via causal graphical models

Haiyang Shi[1,6], Geping Luo[2,3,4,6], Olaf Hellwich[7], Alishir Kurban[2,3,4,6], Philippe De Maeyer[2,3,5,6] and Tim Van de Voorde[5,6]

[1] School of Earth Sciences and Engineering, Hohai University, Nanjing 211100, China.
[2] State Key Laboratory of Desert and Oasis Ecology, Xinjiang Institute of Ecology and Geography, Chinese Academy of Sciences, Urumqi, Xinjiang, 830011, China.
[3] College of Resources and Environment, University of the Chinese Academy of Sciences, 19 (A) Yuquan Road, Beijing, 100049, China.
[4] The National Key Laboratory of Ecological Security and Sustainable Development in Arid Region (proposed), Chinese Academy of Sciences, Urumqi, China.
[5] Department of Geography, Ghent University, Ghent 9000, Belgium.
[6] Sino-Belgian Joint Laboratory of Geo-Information, Ghent, Belgium.
[7] Department of Computer Vision & Remote Sensing, Technische Universität Berlin, 10587 Berlin, Germany.

**Correspondence to:** Geping Luo (luogp@ms.xjb.ac.cn) and Olaf Hellwich (olaf.hellwich@tu-berlin.de)

**Submitted to:** *Biogeosciences*

**Abstract**

Using statistical methods that not directly representing the causality between variables to attribute climate and plant traits to control ecosystem function may lead to biased perceptions. We revisited this issue using a causal graphical model, the Bayesian network (BN), capable of quantifying causality by conditional probability tables. Based on expert knowledge and climate, vegetation, and ecosystem function data from the FLUXNET flux stations, we constructed a BN representing the causal relationship of 'climate-plant trait-ecosystem function'. Based on the sensitivity analysis function of the BN, we attributed the controls of climate and plant traits to ecosystem function and compared the results with those based on Random forests and correlation analysis. The main conclusions of this study include: BN can be used for the quantification of causal relationships between complex ecosystems in response to climate change and enables the analysis of indirect effects among variables. The causality reflected in the BN is as good as the expert knowledge of the causal links. Compared to BN, the feature importance difference between 'mean vapor pressure deficit and cumulative soil water index' and 'maximum leaf area index and maximum vegetation height' reported by Random forests is higher and can be overestimated. With the causality relation between correlated variables constructed, BN-based sensitivity analysis can reduce the uncertainty in quantifying the importance of correlated variables. The understanding of the mechanism of indirect effects of climate variables on ecosystem function through plant traits can be deepened by the chain casuality quantification in BNs.

**1 Introduction**

Ecosystem function is the capacity of natural processes and components to provide goods and services that satisfy human needs, either directly or indirectly (de Groot et al., 2002). Ecosystem functions include the physicochemical and biological processes within the ecosystem to maintain terrestrial life. Terrestrial ecosystems have provided a variety of important ecosystem functions for our society (Manning et al., 2018). Plant traits' role as important determinants of ecosystem functions has been widely recognized (Chapin Iii et al., 2000), and various trait syndromes can result in distinct broad differences in ecosystem functions (Reichstein et al., 2014). In the context of global climate change, it is also essential to understand the potential changes in ecosystem functions (Grimm et al., 2013). The response of terrestrial ecosystem function to changes in climate, plant traits, and the corresponding mechanisms, are complex due to enormous spatial and temporal variations across ecosystems, climate zones, and also space-time scales (Diaz and Cabido, 1997; Madani et al., 2018; Myers-Smith et al., 2019). Given the enormous variations, on the global scale, these issues have not been clarified well.

In the past decades, measurements of ecosystem functions have been increasingly available to support studies of the relations between ecosystem functions and climate variables. For example, eddy-covariance flux tower observations (Baldocchi, 2014) for carbon flux (i.e., net ecosystem exchange (NEE)) and water flux (i.e., evapotranspiration (ET)) have been widely used to investigate changes in ecosystem functions and their responses to climate change, vegetation condition changes, etc (Jung et al., 2020, 2010; Migliavacca et al., 2021; Peaucelle et al., 2019). With the increase in such observations, various statistical analysis approaches such as machine learning (Barnes et al., 2021; Migliavacca et al., 2021; Reichstein et al., 2019; Shi et al., 2022b, a;

Tramontana et al., 2016) have been used to mine the hidden information on the effects of climate change and its
induced changes in vegetation, etc. on ecosystem function variables such as carbon and water flux, which has
not been understood in depth by process-based models (e.g., biogeochemistry models (Sakschewski et al.,
2016)). For example, using Random forests (RF) and principal component analysis (PCA), a recent study
(Migliavacca et al., 2021) quantified the three main axes of terrestrial ecosystem function and their drivers based
on observations of carbon and water fluxes of FLUXNET stations (Pastorello et al., 2020) and various climate
and plant trait variables. Generally, data-driven approaches have become increasingly important recently in this
area (Reichstein et al., 2019).

However, compared to the process-based models, most of these data-driven approaches lack representation of
the causality and detailed processes in the relations between ecosystem function and climate, despite the widely
recognized complex causal interactions between ecosystems and climate systems (Reichstein et al., 2014).
Conventional methods such as multiple linear regression have been questioned in attribution studies of the
relationship between climate and the carbon cycle (Wang et al., 2022). For example, the use of multiple linear
regression may underestimate the direct effect of soil moisture possibly due to the covariance between variables
(Wang et al., 2022). For machine learning techniques, current common algorithms such as RF (Migliavacca et
al., 2021) can report the importance of features (IMP) to measure their contributions to the prediction model.
However, IMP-based attribution to the target variable can also be unreliable if considerable confounders and
correlations between predictor variables exist (Strobl et al., 2008; Toloşi and Lengauer, 2011). The less relevant
predictors can replace the predictive predictors (due to correlation) and thus receive undeserved high feature
importance (Strobl et al., 2008). Correlations between predictors can lead to biased IMP-based findings. It is
thus important to recognize the difference between correlation and causality in these approaches and represent
detailed causal relations between features, rather than the unreliable IMP rankings generated from correlated
features.

Bayesian network (BN) is a causal graphical model based on conditional probability representation (Friedman et
al., 1997; Pearl, 1985) that characterizes the transmission of cause and effect through conditional probabilities
between variables. Currently, BN has been used in modeling causal relationships in many fields and has
demonstrated advantages in causal interpretation, including in the fields such as hydrology and ecology (Chan et
al., 2010; Keshtkar et al., 2013; Milns et al., 2010; Pollino et al., 2007; Shi et al., 2021a, b; Trifonova et al.,
2015). However, BN has rarely been used in the study of the attribution of changes in ecosystem function.
Therefore, this study used BN to attribute the controls of climate and plant traits to ecosystem function by
quantifying the causal relationships involved. The data used was from a previous study (Migliavacca et al.,
2021) which extracted ecosystem function, climate, and plant trait variables for FLUXNET flux stations. The
construction of the causal structure of BN referred to the previous expert knowledge of this system (Reichstein
et al., 2014). Further, by comparing BN-based attribution analysis, linear correlation analysis, and RF-based
IMP reported by the previous study (Migliavacca et al., 2021), we investigated the adding-values of using BN
for causal analysis and discussed its prospects in this paper.
**2 Methodology**
**2.1 Data**
The used variables (Table 1) include the carbon and water fluxes of the FLUXNET flux tower sites and the
ecosystem function variables derived from them, and information on the corresponding climate variables as well
as plant traits:
a)    Ecosystem function variables: underlying Water Use Efficiency (uWUE), maximum evapotranspiration

(ETmax), maximum surface conductance (GSmax), maximum net $CO_2$ uptake of the ecosystem

(NEPmax), Gross Primary Productivity at light saturation (GPPsat), Mean basal ecosystem respiration at a

reference temperature of 15 °C (Rb), and apparent carbon-use efficiency (aCUE).

b)    Plant trait variables: ecosystem scale foliar nitrogen concentration (Nmass), Maximum Leaf Area Index

(LAImax), Maximum vegetation height (Hc). Of the total 202 sites (Migliavacca and Musavi, 2021), 101

sites have Nmass data, 153 sites have LAImax data, and 199 sites have Hc data. Only 98 have data on all

these three plant trait variables.

c)    Climate variables: mean incoming shortwave radiation (SWin), Mean temperature (Tair), Mean Vapor

Pressure Deficit (VPD), Mean annual precipitation (P), and cumulative soil water index (CSWI).


These data have different producing processes, including those calculated from flux data, site records, extracted
from remote sensing data, etc. The detailed calculation methods can be found in Migliavacca et al., 2021.

Table 1. The variables used and the discretization of their values in BN.

| Variable node | Definition and units | Type | Approach (Migliavacca et al., 2021) | Discretization in BN (equal quantile thresholds: 0%, 33.33%, 66.67%, and 100% percentile values) |
|---|---|---|---|---|
| uWUE | underlying Water Use Efficiency [gC kPa^0.5 kgH$_2$O$^{-1}$] | Ecosystem function | It was calculated from GPP, VPD, and ET (Zhou et al., 2014). The median of the half-hourly retained uWUE values was used for each site. It was further filtered by the following conditions: (i) SWin > 200 W m$^{-2}$; (ii) no precipitation event for the last 24 hours, when precipitation data are available; and (iii) during the growing season: daily GPP > 30% of its seasonal amplitude. | 0.068, 2.51, 3.18, 5.332 |
| ETmax | maximum evapotranspiration in the growing season [mm] | Ecosystem function | ETmax was computed as the 95th percentile of ET in the growing season. It was also filtered by the same filtering applied to the uWUE calculation. | 0.059, 0.17, 0.23, 0.423 |
| GSmax | maximum surface conductance [m s$^{-1}$] | Ecosystem function | GSmax was computed by inverting the Penman-Monteith equation after calculating the aerodynamic conductance. The 90th percentile of the half-hourly GS of each site was calculated and used as the GSmax of each site. | 0.0013, 0.0077, 0.0123, 0.0566 |

| NEPmax | maximum net CO2 uptake of the ecosystem [umol $CO_2$ $m^{-2}$ $s^{-1}$] | Ecosystem function | NEPmax was computed as the 90th percentile of the half-hourly net ecosystem production in the growing season (when daily GPP is > 30% of the GPP amplitude). | 1.953, 15.3, 24.4, 42.82 |
|---|---|---|---|---|
| GPPsat | Gross Primary Productivity at light saturation [umol $CO_2$ $m^{-2}$ $s^{-1}$] | Ecosystem function | GPPsat was computed as the 90th percentile estimated from half-hourly data by fitting the hyperbolic light response curves. The 90th percentile from the GPPsat estimates of each site was extracted. | 3.042, 17.49, 27.74, 47.6 |
| Rb | Mean basal ecosystem respiration at a reference temperature of 15 °C [umol $CO_2$ $m^{-2}$ $s^{-1}$] | Ecosystem function | Rb was derived from night-time NEE measurements. For each site, the mean of the daily Rb value was computed. | 0.144, 2.07, 3.12, 10.67 |
| aCUE | apparent carbon-use efficiency | Ecosystem function | aCUE was calculated by aCUE = 1- (Rb/GPP) and the median value of daily aCUE is used. | -1.19, 0.4, 0.74, 1 |
| Nmass | ecosystem scale foliar nitrogen concentration [gN 100 $g^{-1}$] | Plant trait | Nmass was computed as the community-weighted average of foliar N% of the major species at the site sampled at the peak of the growing season or gathered from the literature (Musavi et al., 2016, 2015; Fleischer et al., 2015; Flechard et al., 2020). | 0.65, 1.15, 1.76, 4.44 |
| LAImax | Maximum Leaf Area Index [$m^2$ $m^{-2}$] | Plant trait | LAImax was collected from the literature (Migliavacca et al., 2011; Flechard et al., 2020), the FLUXNET Biological Ancillary Data Management (BADM) product, and/or site principal investigators. | 0.17, 2.27, 4.5, 12.9 |
| Hc | Maximum vegetation height [m] | Plant trait | Hc was collected from the literature (Migliavacca et al., 2011; Flechard et al., 2020), the BADM product, and/or site principal investigators. | 0.04, 1.7, 16.0, 80.1 |
| SWin | Mean incoming shortwave radiation [W $m^{-2}$] | Climate | SWin was from FLUXNET data. | 54.43, 134.18, 182.44, 266.04 |
| Tair | Mean temperature [degree C] | Climate | Tair was from FLUXNET data. | -10.45, 6.62, 14.73, 28.1 |
| VPD | Mean Vapor Pressure Deficit [hPa] | Climate | VPD was from FLUXNET data. | 0.62, 3.38, 5.76, 26.08 |
| P | Mean annual precipitation [cm/year] | Climate | P was from FLUXNET data. | 5.51, 45.28, 79.29, 256.61 |

| CSWI | cumulative soil water index | Climate-related soil water availability | CSWI was computed as a measure of water availability (Nelson et al., 2018). | -93.49, -1.24, 2.01, 4.47 |
|---|---|---|---|---|


## 2.2 BN for analyzing causal relations

### 2.2.1 BN structures

Based on expert knowledge (Reichstein et al., 2014), we constructed the structure of BN containing the causal relationships between plant traits and ecosystem function variables: 'BN_plant_trait'. The causal links between the variables were referred to the relationship diagram in the upper part of Figure 1. Further, we added the climate variables and the corresponding causal relationships, expanding 'BN_plant_trait' to 'BN_plant_trait_climate', which further incorporates the climate variables and their impacts on the system (Figure 1). The explanation of added causal links was shown in Table 2.

Each node is discretized for the BN compiling by the software Netica. The equal quantile (Nojavan A. et al., 2017) three-level discretization (the distribution of nodes (Figure S1) is divided into three levels) for each node is applied by the discretization thresholds of 0%, 33.33%, 66.67%, and 100% percentile values of the data distribution (Table 1) given the limitation of the amount of training data.

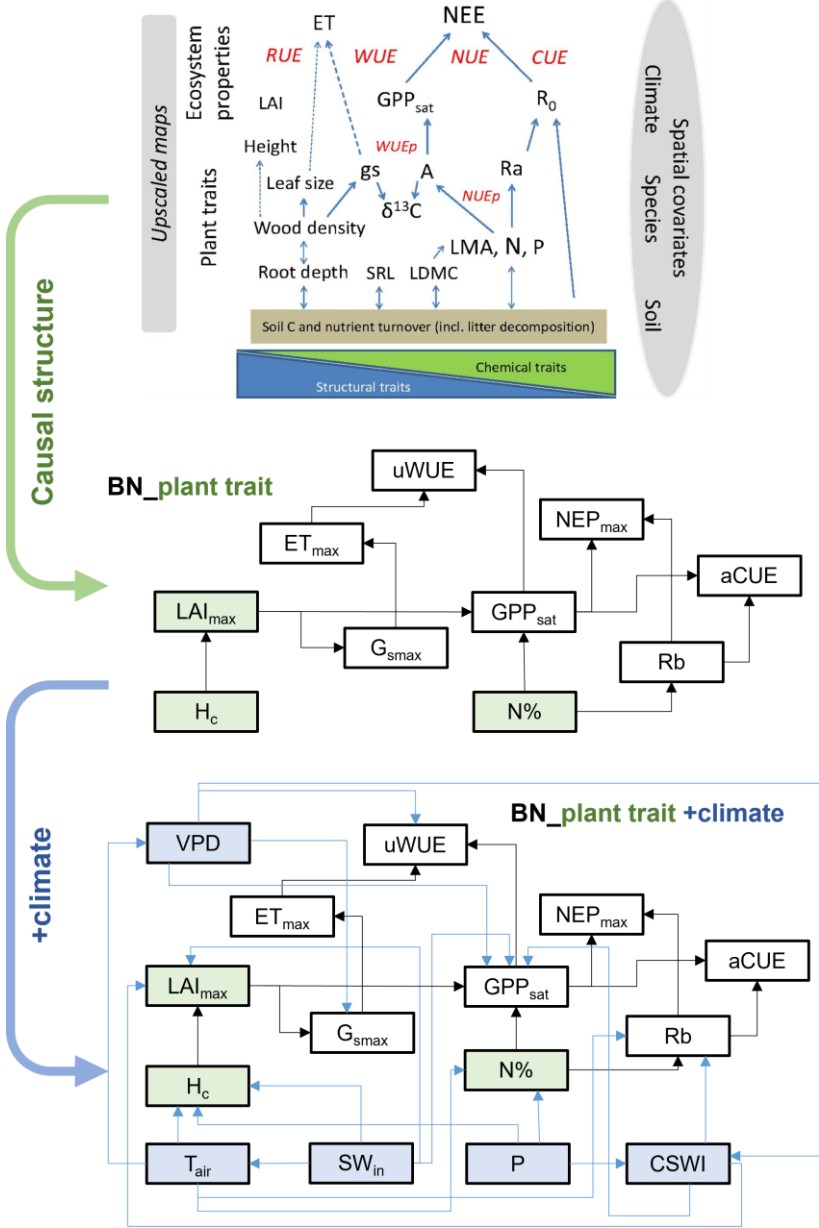


Figure 1. The structure of two Bayesian networks (BNs) for attribution of variations in ecosystem functions.
'BN_plant_trait' in the median part incorporated the causal effects of plant traits (box in slight green) on
ecosystem functions (box in white) from expert knowledge as the relation diagram on the upper part (Reichstein
et al., 2014). 'BN_plant_trait_climate' in the lower part further incorporated the causal impacts of climate
variables (box in light blue).

Table 2. Explanation of the added causal links between climate variable nodes, plant trait nodes, and ecosystem
function variable nodes in the BNs.

| Casual links | | Explanation | References |
|---|---|---|---|
| Parent node | Child node | | |

| | | | |
|------|--------|------------------------------------------------------------|---------------------------------------------------------|
| VPD | uWUE | uWUE= GPP· $VPD^{0.5}$/ET | (Zhou et al., 2014) |
| VPD | GSmax | stomatal and surface conductance declines under an increase in VPD | (Grossiord et al., 2020; Wever et al., 2002) |
| VPD | GPPsat | leaf and canopy photosynthetic rates decline when atmospheric VPD increases due to stomatal closure | (Yuan et al., 2019; Konings et al., 2017) |
| VPD | CSWI | CSWI declines under an increase in VPD | (Nelson et al., 2018) |
| Tair | VPD | higher air temperature corresponds to higher saturated water vapor pressure and can drive an increase in VPD | (Yuan et al., 2019) |
| Tair | Hc | the temperature limitation on canopy height variation | (Moles et al., 2009) |
| Tair | Nmass | increase in air temperature may decrease plant nitrogen concentration and leaf nitrogen content. | (Weih and Karlsson, 2001; Reich and Oleksyn, 2004) |
| Tair | Rb | temperature strongly influences Rb through the laws of thermodynamics | (Davidson and Janssens, 2006; Enquist et al., 2003; Brown et al., 2004) |
| SWin | LAImax | solar radiation affects vegetation conditions and phenology | (Günter et al., 2008; Liu et al., 2016; Borchert et al., 2015; Wagner et al., 2017) |
| SWin | Hc | solar radiation affects the distribution and composition of ecosystems through photosynthesis and the water cycle | (Borchert et al., 2015; Guisan and Zimmermann, 2000; Piedallu and Gégout, 2007) |
| SWin | GPPsat | solar radiation affects ecosystem productivity and plant growth | (Monteith, 1972; Borchert et al., 2015; Guisan and Zimmermann, 2000) |
| P | Hc | the hydraulic limitation hypothesis on canopy height variation | (Moles et al., 2009; Ryan and Yoder, 1997; Koch et al., 2004) |
| P | Nmass | leaf nitrogen concentration per unit mass may decrease with increasing precipitation | (Santiago and Mulkey, 2005; Wright and Westoby, 2002) |
| P | CSWI | CSWI declines under a decrease in P | (Nelson et al., 2018) |
| CSWI | LAImax | soil moisture affects vegetation conditions | (Patanè, 2011) |
| CSWI | Rb | soil moisture affects the temperature dependence of ecosystem respiration | (Xu et al., 2004; Flanagan and Johnson, 2005; Wen et al., 2006) |
| CSWI | GPPsat | soil moisture can reduce GPP through ecosystem water stress | (Green et al., 2019) |


**2.2.2 BN evaluation and node sensitivity analysis**

Based on the Bayesian network (BN), the joint impacts of multiple variables and their causal relations are analyzed. A BN can be represented by nodes $X_1$, $X_2$, $X_3$ to $X_n$ and the joint distribution (Pearl, 1985):

$$Pa(X) = Pa(X_1, X_2, ..., X_n) = \prod_{i=1}^{n} Pa(X_i | pa(X_i)) \tag{1}$$

where $pa(X_i)$ is the probability of the parent node $X_i$. Expectation-maximization (Moon, 1996) is used to address the data with missing values and then compile the BN.

We used k-fold cross-validation to verify the reliability of the BN. The k-fold approach has been widely used in previous studies for the validation of BNs (Marcot, 2012). In this study, k is set as 10 as commonly used (Marcot and Hanea, 2021). We choose ETmax, GPPsat, and NEPmax for cross-validation of accuracy, and the predicted status (status with the highest probability bar value) of the nodes will be compared with the actual status and the classification accuracy will be calculated. These three nodes are the main terminal nodes and primary objectives of the BN and represent the main water and carbon-related ecosystem functions, respectively. The accuracy of these three variables can largely reflect the overall performance of BN.

Sensitivity analysis is used for the evaluation of the strength of the causal relations between nodes based on mutual information (MI). MI is calculated as the entropy reduction of the child node resulting from changes found at the parent node (Shi et al., 2020):

$$MI = H(Q) - H(Q|F) = \sum_q \sum_f P(q, f) \log_2 \left( \frac{P(q,f)}{P(q)P(f)} \right) \tag{2}$$

where H represents the entropy, Q represents the target node, F represents the set of other nodes and q and f represent the status of Q and F. In this study, we assessed the sensitivity of ecosystem function variables to climate and plant trait variables.

**2.2.3 Comparing different approaches used for attribution analysis**

Further, to clarify the adding-values of considering causality in the attribution analysis of controls on ecosystem functions, the results of the BN-based sensitivity analysis (BN_sens) were compared with the other two approaches. They are the results of the absolute values of additional linear correlation analysis (linear_corr) in this study and the findings in Migliavacca et al., 2021 using RF feature importance (RF_imp). BN_sens and linear_corr directly measure the effects of plant traits and climate variables on ecosystem function variables, while RF_imp measures their effects on the three principal components (PC1, PC2, and PC3) of ecosystem function variables, which were reported as the three major axes of ecosystem function by Migliavacca et al., 2021. It was obtained from principal component analysis of 12 ecosystem function variables which included the six variables uWUE, ETmax, GSmax, NEPmax, GPPsat, and Rb used in the methods BN_sens and linear_corr. The first axis (PC1) explains 39.3% of the variance and is dominated by maximum ecosystem productivity properties, as indicated by the loadings of GPPsat and NEPmax, and maximum evapotranspiration (ETmax). The second axis (PC2) explains 21.4% of the variance and refers to water-use strategies as shown by the loadings of water-use efficiency metrics, evaporative fraction, and GSmax. The third axis (PC3) explains 11.1% of the variance and includes key attributes that reflect the carbon-use efficiency of ecosystems. PC3 is dominated by apparent carbon-use efficiency, basal ecosystem respiration (Rb), and the amplitude of evaporative fraction (Migliavacca et al., 2021).

## 3 Results

### 3.1 Correlation analysis

Linear correlation analysis of the variables (Figure 2) showed significant (P < 0.05) linear correlations between the ecosystem function variables and some of the climate and plant trait variables. SWin and VPD showed negative correlations with these ecosystem function variables. LAImax/ Hc showed significant positive relationships with most of the ecosystem function variables and significant negative relationships with SWin and VPD. Nmass only showed a positive relationship with ETmax. In addition, the majority of the ecosystem function variables showed significant (P < 0.05) positive correlations with each other.

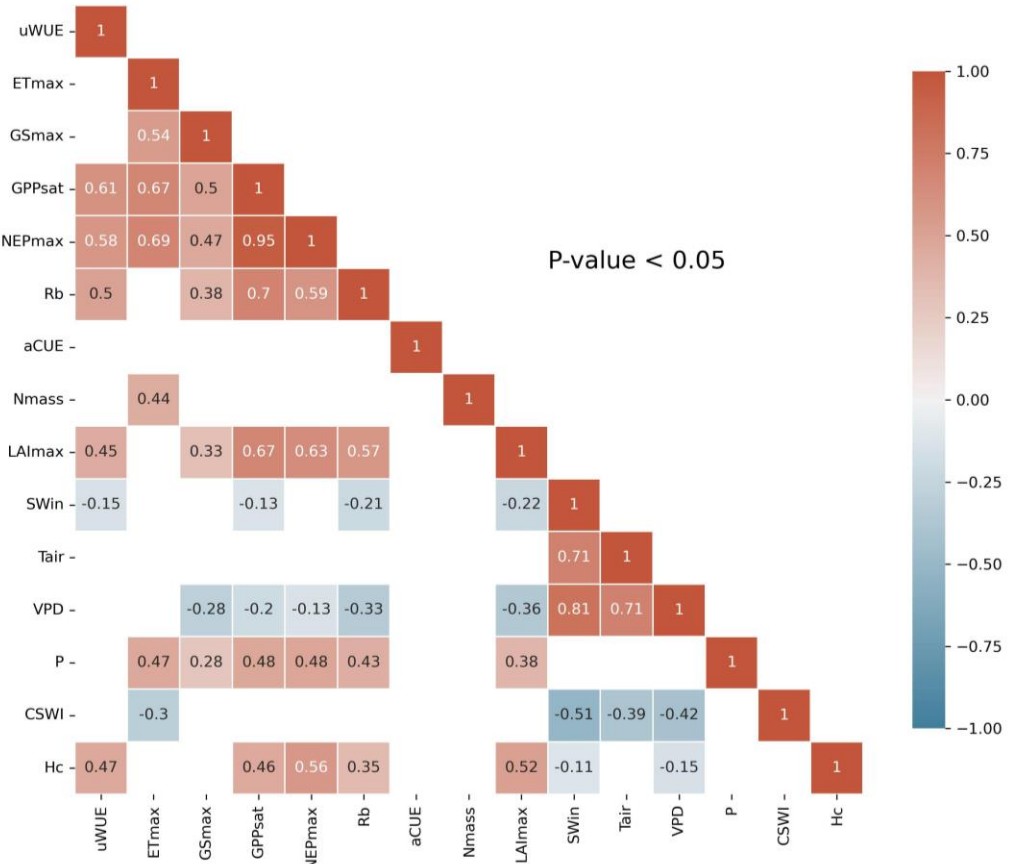

Figure 2. Correlation coefficient matrix of ecosystem functions and climate and plant trait variables for FLUXNET sites. Only correlation coefficients with p-values less than 0.05 level of significance is shown.

### 3.2 BN-based analysis

We compiled two different BNs (i.e., BN_plant_trait and BN_plant_trait_climate) (Figure 3) and found that the probability distributions of the values of the common nodes (ecosystem function and plant trait variable nodes) differed a little (e.g., in the probability distribution of LAImax, Hc, and Nmass) between the two BNs. Compared to BN_plant_trait, in BN_plant_trait_climate, the climate variables of sites with missing plant trait data forced the changes in the probability distributions of LAImax, Hc, and Nmass. In the EM algorithm, for

sites with missing plant trait data, existing relationships (obtained from observations from other sites) between plant trait variables and climate variables are used in the data interpolation of plant trait variables. In BN_plant_trait_climate, the added linkages of climate variables to plant trait variables resulted in higher probability values of the low-value status of the plant trait variables.

The 10-fold cross-validation of the nodes ETmax, GPPsat, and NEPmax showed relatively high accuracy. The classification accuracy (Table S1) of the status of ETmax was 60.9%, the classification accuracy of the status of NEPmax was 84.2% and the classification accuracy of the status of GPPsat was 75.2%.

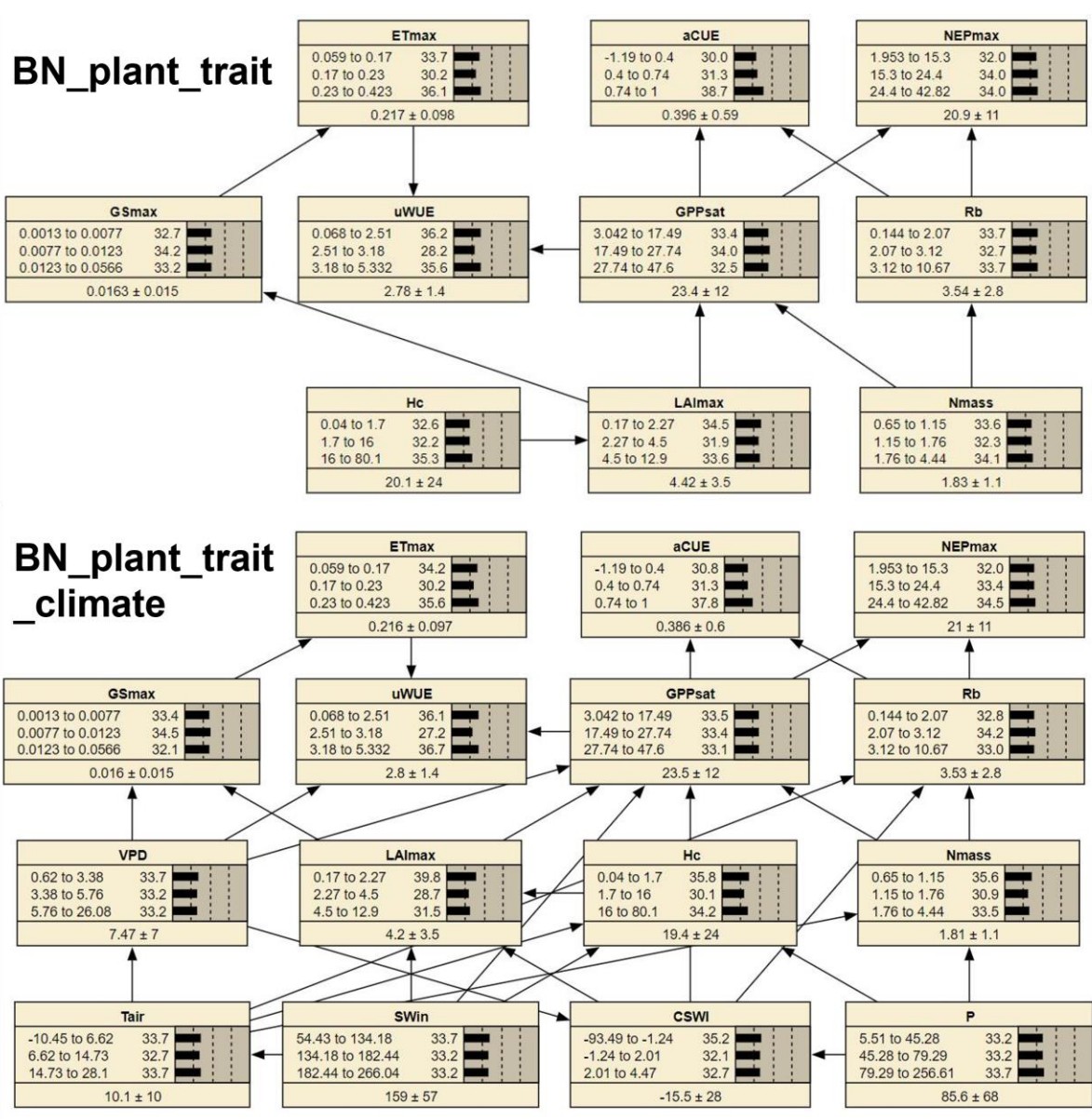

Figure 3. The compiled two BNs ('BN_plant_trait' and 'BN_plant_trait_climate'). The bars of each node represent its probability distribution. At the bottom part of each node, the left and right side values of the '±' are the mean and standard deviation of the distribution, respectively.

We performed sensitivity analyses (Figure 4) on the ecosystem function variables in both BNs to assess their
sensitivity to various climate and plant trait variables. We also calculated the difference in sensitivity MI
between the two BNs (Figure 4) to compare the change in sensitivity of ecosystem function to each variable
after adding further climate variables to the plant trait variables only. The sensitivity of different ecosystem
function variables to plant traits and climate variables was highly variable in both BNs. The magnitude of
sensitivity of ecosystem function nodes to plant traits and climate variables was related to whether these plant
traits and climate variables were set as their parent nodes. In BN_plant_trait, for the carbon fluxes GPPsat and
NEPmax, Nmass, and LAImax had higher sensitivity due to Nmass and LAI being set as their parent nodes. For
the water flux ETmax, it does not have high sensitivity to plant trait variables such as LAImax and Hc, although
these plant trait variables are set as the parent nodes of ETmax. This indicates the difference in the strength of
the control effects of plant traits on carbon and water fluxes.

In the sensitivity analysis of BN_plant_trait_climate, the sensitivity patterns of the ecosystem function variables
changed as a result of the inclusion of climate variables and the change in causality they introduced. The
sensitivity of the ecosystem function variables to climate variables was significantly increased (especially for
Tair, VPD, and CSWI). The control of plant traits on ecosystem function in BN_plant_trait is also partially
transformed into an indirect effect of climate variables by first controlling plant trait variables and then
controlling ecosystem function. For example, in BN_plant_trait_climate, for GPPsat, a decrease in the
sensitivity of GPPsat to LAImax and an increase in the sensitivity to Tair was observed after the causal chain of
Tair influencing Hc, LAImax, and then GPPsat was set. This can be explained by the fact that higher
temperatures promote vegetation growth and thus may increase LAImax, which then indirectly alters the
probability distribution of the GPPsat node. In previous studies based on statistical methods that did not consider
the chain causality, this indirect control on GPPsat from Tair may have been included in the contribution of
LAImax to GPPsat. Similarly, a chain causality of P by first affecting Nmass and then indirectly GPPsat was
also found. However, the effect of P by first affecting Hc, LAImax, and then indirectly affecting ETmax and
GSmax appears to be not large.

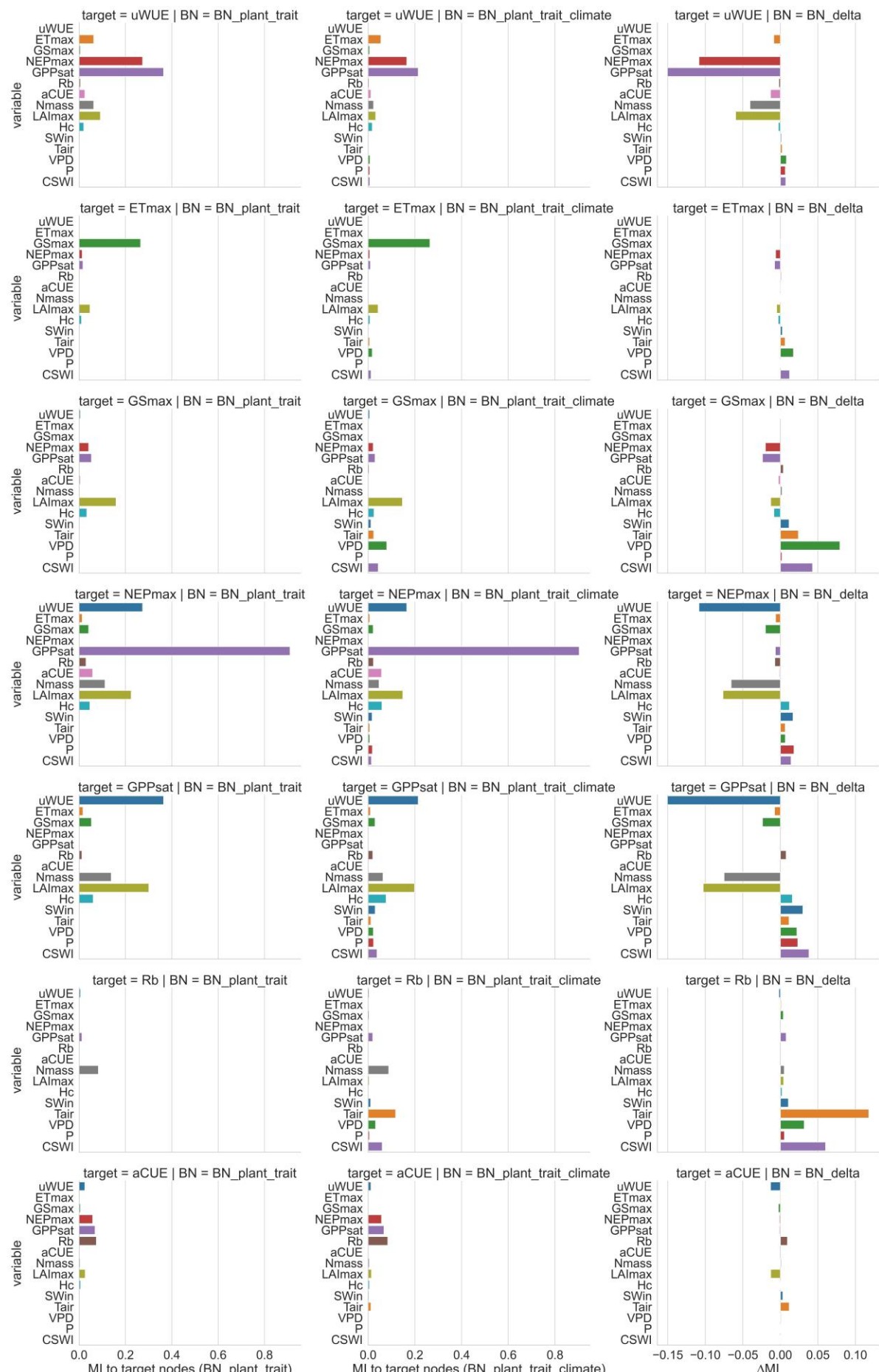


Figure 4. Sensitivity of ecosystem function variables to other variables in different networks based on mutual information (MI). The left column is the sensitivity analysis of BN_plant_trait, the middle column is the sensitivity analysis of BN_plant_trait_climate, and the right column is the difference between the reported sensitivity of BN_plant_trait_climate and the sensitivity of BN_plant_trait. For BN_plant_trait, the MI values of climate variables to ecosystem function variables are all 0 because they do not contain climate variables. For each ecosystem function in these two BNs, its sensitivity to its child node is not shown (set as 0) because child nodes are not considered causal variables and thus are not evaluated in the attribution.

**3.3 Comparing results from RF-based, BN-based analysis, and correlation analysis**

All three methods show the importance of the plant trait variables in explaining the variation of various ecosystem function variables (Figure 5). LAImax was the most important of the three methods in explaining the variation of maximum ecosystem productivity properties (corresponding to PC1). In contrast to the results of the other two methods, in linear_corr, SWin and VPD were the least important, while P was more important. Comparing RF_imp and BN_sens, the overall pattern of importance is similar, but there are differences. For water-use strategies (corresponding to PC2), Hc is ranked first and LAI last in RF_imp, but in BN_sens, LAI is slightly more important than Hc. In linear_corr, Hc and LAI are of similar importance. For PC3, VPD ranks first and is more important than Tair in RF_imp. But in BN_sens, Tair is more important than VPD. Among the three moisture-related climate variables (i.e., VPD, P, and CSWI), CSWI appears to be the least important in RF_imp but is comparable to VPD in BN_sens.

Given the limitations of RF_imp in responding to the correlated variables (Strobl et al., 2008), the difference between the significance of VPD and CSWI reported by RF_imp may be overestimated. For the ecosystem functions related to water-use strategies, the difference between LAImax and Hc reported by BN_sens is also much smaller than the difference reported by RF_imp. It implied that, with the causality relation between correlated variables constructed, BN_sens reduced the uncertainty in quantifying the importance of correlated variables.

| | Methods | Nmass | LAImax | Hc | SWin | Tair | VPD | P | CSWI |
|---|---|---|---|---|---|---|---|---|---|
| PC1 | RF_imp | 10.80% | 16.60% | 14.50% | 7.60% | 9.10% | 11.70% | 6.70% | 4.00% |
| PC2 | RF_imp | 5.10% | 4.50% | 14.90% | 10.70% | 11.20% | 7.40% | 9.00% | 8.30% |
| PC3 | RF_imp | 7.00% | 2.80% | 5.40% | 9.30% | 8.00% | 15.40% | 6.50% | 4.90% |
| | | | | | | | | | |
| GPPsat | BN_sens | 0.0635 | 0.1980 | 0.0766 | 0.0299 | 0.0116 | 0.0221 | 0.0232 | 0.0380 |
| NEPmax | BN_sens | 0.0464 | 0.1482 | 0.0588 | 0.0168 | 0.0064 | 0.0065 | 0.0181 | 0.0142 |
| ETmax | BN_sens | 0.0006 | 0.0424 | 0.0076 | 0.0028 | 0.0063 | 0.0174 | 0.0006 | 0.0122 |
| uWUE | BN_sens | 0.0228 | 0.0321 | 0.0174 | 0.0012 | 0.0023 | 0.0080 | 0.0066 | 0.0072 |
| GSmax | BN_sens | 0.0022 | 0.1464 | 0.0246 | 0.0115 | 0.0239 | 0.0793 | 0.0019 | 0.0429 |
| Rb | BN_sens | 0.0880 | 0.0043 | 0.0021 | 0.0106 | 0.1177 | 0.0317 | 0.0053 | 0.0602 |
| aCUE | BN_sens | 0.0049 | 0.0138 | 0.0056 | 0.0033 | 0.0117 | 0.0009 | 0.0004 | 0.0007 |
| | | | | | | | | | |
| GPPsat | linear_corr | | 0.67 | 0.46 | 0.13 | | 0.20 | 0.48 | |
| NEPmax | linear_corr | | 0.63 | 0.56 | | | 0.13 | 0.48 | |
| ETmax | linear_corr | 0.44 | | | | | | 0.47 | 0.30 |
| uWUE | linear_corr | | 0.45 | 0.47 | 0.15 | | | | |
| GSmax | linear_corr | | | | | | 0.28 | | |
| Rb | linear_corr | | 0.57 | 0.35 | 0.21 | | 0.33 | 0.43 | |
| aCUE | linear_corr | | | | | | | | |

Figure 5. Comparisons of relationships of ecosystem functional variables to plant traits and climate variables in different analyses. Method RF_imp is Random forest variable importance (Migliavacca et al., 2021) (see Methodology section). Method linear_corr is Linear correlation analysis with the absolute values of Pearson correlation coefficients (see Methodology section). Method BN_sens is a BN-based sensitivity analysis with sensitivity values MI reported. The values in each method group are in red for high values and in blue for low values. The color depth is dependent on values and the scale is the same in each row.

## 4 Discussions

Based on BN, this study investigates the prospect of using causal graphical models to revisit and attribute the control of climate and plant trait variations to ecosystem functions. Because of the inclusion of the constraints provided by expert knowledge (Reichstein et al., 2014) and other perceptions from many previous studies, BN-based attribution analysis is relatively reliable in terms of the represented mechanisms of causal links. It can update our knowledge of the contribution of some teleconnection variables through causal chains. The effective implementation of BN-based causal analysis may depend on the reliability of the causal relationships provided by expert knowledge (directional links between variables). We can establish the connection relationships and network structures between variables from expert knowledge and assign the specific quantification of the

connection relationships (conditional probability tables) to observations (Shi et al., 2021a). If further combined
with findings from process-based models, it is promising to significantly improve our understanding of the
complex 'climate-plant trait-ecosystem function' relationships by comparing detailed relationships and structural
influences between variables.

BN essentially factorized the joint probability distribution between various variables into a series of conditional
probability distributions (Ramazi et al., 2021), and the reliability of this approach relied on the setting of causal
control relationships between nodes. Expert knowledge was thus critical in the construction of BNs, especially
when modeling complex systems. In addition to the causal relationship between nodes, the meaning represented
by each node, the data source/ approach, and the spatial and temporal resolution may also have impacts on the
results. For example, in this study, for multiple water use efficiency-related variables in Migliavacca et al., 2021,
uWUE was chosen, and for Rb, the mean value of Rb was chosen. The results of BN-based analysis may vary if
different representations or meanings of nodes are selected. The way the data of each variable is observed/
produced, the spatial and temporal resolution of the data, etc. can also affect the understanding of the role of
these variables in the data-driven BN. Some variables may be very important in the attribution of actual
ecosystem function variation, but their importance may be underestimated due to limitations in the inherent
observational accuracy of their data, and differences in their spatial and temporal scales from other variables. In
addition, some variables such as soil moisture may be difficult to obtain due to the lack of continuous site-scale
long-term observations. Using the water balance method to calculate CSWI as a proxy may introduce errors.
Since the CSWI calculation method relies on P, etc., the obtained relationship between P, CSWI, and other nodes
may have contained empirical components. If the availability of measurements of some nodes is low, modelers
should be cautious about the empirical dependencies with other nodes that may be included in the alternative
data approaches. Thus, the alternative use of multiple derivatives of a variable and data generated by different
methods for the construction of different BNs can help us to recognize how the uncertainty in the nodes and data
can influence BN-based attribution findings. Different node discretization schemes may also affect the
conditional probability table between nodes as well as the sensitivity (Nojavan A. et al., 2017). Other alternative
discretization schemes with the commonly used three levels may also be effective, such as using 'mean-std'
(mean minus 1 standard deviation) and 'mean+std' (mean plus 1 standard deviation) as discretization thresholds,
which will result in a change in the relationship between BN nodes. And further if extreme values such as 5th
and 95th pencentile are used in the node value discretization, it may be beneficial on quantifying the causal
control of extreme conditions of nodes on other nodes.

When considering higher-order effects (Bairey et al., 2016), the relationships between plant traits, climate
variables, and ecosystem function variables can be very complex. One variable may affect the relationship
between two other variables rather than directly affecting these two variables (Bairey et al., 2016). BN may have
limitations in directly analyzing such higher-order effects because BN requires the modeler to explicitly set
direct causal relationships between nodes. To analyze the higher-order effects, we can add nodes that directly
represent the relationship between the variables. For example, the correlation coefficient of two variables can be
used as a node and this node is connected to other nodes in the BN so that the control effect of other nodes on
this correlation coefficient can be explored. Such implements may be useful to deepen the impact of various
higher order effects.
Besides, the BN in this study was mainly based on data averaged over multiple years, thus possibly partially
underestimating the effect of temporal variations in the relationships between variables. Another limitation of
the BN proposed above is that the causal relationships between variables are unidirectional, while it is difficult
to represent interactions and feedback between variables (Marcot and Penman, 2019). In future studies, to
address these two issues, BN based on temporal dynamics can be promising (Figure 6). By refining the
interaction of temporal lags between variables, it is possible to incorporate not only temporal variation but also
control factors that attribute interactions and feedback between variables. For example, the interaction and
feedback mechanisms of VPD, soil moisture, and ET with lag effects (Figure 6) and their impacts on ecosystems
have attracted extensive interest from researchers (Anderegg et al., 2019; Humphrey et al., 2021; Lansu et al.,
2020; Liu et al., 2020; Xu et al., 2022; Zhou et al., 2019), but conventional statistical methods have been
ineffective in analyzing such relationships with both interactive causality and temporal lags. In contrast, the BN
proposed here, which incorporates feedback effects and lagged effects that were common in climate-ecosystem
relations (Lin et al., 2019), is potentially able to address this issue from a data-driven approach. In the practical
modeling, different periods of the same node may still be not independent. Therefore, the split scheme of such
periods may be critical. For example, a period between two precipitation events can be treated as one sample,
which can enhance independence between periods. Subsequently, a such period can be divided into smaller
periods such as t, t-1, t-2, etc. to aggregate the node values to appropriate time scales. Thus one sample can
represent the interaction relationship between variables with lags in this period. Finally, we can integrate records
of such periods between two precipitation events from sites across different climate zones and biomes to build
synthesis models for global analysis of such problems. Such research frameworks in BN-based modeling may
be difficult due to high computational costs given the large amount of data. Fortunately, recently proposed new
causal models have the potential to address this limitation, such as the introduction of causality into deep
learning frameworks (Luo et al., 2020; Cui and Athey, 2022). If further combined with the findings of process-
based models, our understanding of climate and ecosystem interactions and feedback and their mechanisms in
time is hopefully deepened.

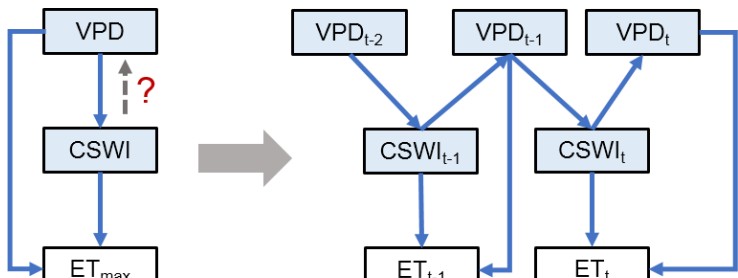

Figure 6. The future BNs with the temporal causality further considered addressing the causality of the
interaction between variables. The VPD-CSWI-ET relationship is used here as an example. t, t-1, and t-2 denote
the current period, the last period, and the period before the last period, respectively. The network on the left
only considers the effect of VPD on CSWI without considering the feedback of CSWI on the VPD. The network
on the right characterizes the VPD-CSWI interaction with the feedback from CSWI at period t-1 to VPD at
period t.
**5 Conclusion**
Based on BN, we revisited and attributed the contribution of climate and plant traits to global terrestrial
ecosystem function. The major conclusions of this study include:
1. BN can be used for the quantification of causal relationships between complex ecosystems in response to
climate change and enables the analysis of indirect effects among variables.
2. Compared to BN, the feature importance difference between 'VPD and CSWI' and 'LAImax and Hc'
reported by Random forests is higher and can be overestimated.
3. With the causality relation between correlated variables constructed, BN_sens can reduce the uncertainty in
quantifying the importance of correlated variables.
4. The understanding of the mechanism of indirect effects of climate variables on ecosystem function through
plant traits can be deepened by the chain casuality quantification in BNs.

**Acknowledgements**
We are grateful to the two anonymous reviewers for their thorough and careful review that led to substantial
improvements in the manuscript. We are also grateful to Dr. Kirsten Thonicke for being the associate editor.
**Financial support**
This research was supported by the Tianshan Talent Cultivation (Grant No. 2022TSYCLJ0001), the Key
projects of the Natural Science Foundation of Xinjiang Autonomous Region (Grant No. 2022D01D01), the
Strategic Priority Research Program of the Chinese Academy of Sciences (Grant No. XDA20060302), and
High-End Foreign Experts Project.
**Author Contributions**
HS and GL initiated this research and were responsible for the integrity of the work as a whole. HS performed
formal analysis and calculations and drafted the manuscript. HS was responsible for the data collection and
analysis. GL, PDM, TVdV, OH, and AK contributed resources and financial support.
**Competing interests**
The authors declare that they have no conflict of interest.
**Code availability**
The codes that were used for all analyses are available from the first author (shihaiyang16@mails.ucas.ac.cn)
upon request.
**Data availability**
The data used in this study can be accessed by contacting the first author (shihaiyang16@mails.ucas.ac.cn) upon
request.

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
