# Peer review of "Revisiting and attributing the global controls on terrestrial"

_Biogeosciences, 2022_

## Author Response (AR1)

**Response to Referee #1**

**1.     General comments**

The authors attempt to build causal links between plant traits, climate and ecosystem functions by constructing a Bayesian Network (BN), where links between traits and functions are based on expert knowledge, while the climatic variables are informed by the model. The authors then reevaluate the relative importance of plant traits and climate in determining ecosystem functions through a sensitivity analysis based mainly on FLUXNET data. Building on this they argue that climate indirectly affects ecosystem functions via its control on plant traits. We agree that, from an ecological perspective and considering the increasing availability of data, exploring different methods to analyze the interactions climate-vegetation involved in the ecosystem functions is a relevant and meaningful research topic. However, the paper is missing an appropriate model validation (making it difficult to evaluate the robustness of the results) and suffers from reproducibility issues as some important methodological points and choices require clarification or additional information. We start by providing some major concerns followed by minor comments in order of appearance.

Response: Thank you for your insightful comments, which have been very helpful in improving the manuscript. This manuscript will be revised in accordance with your comments. In terms of model validation, we will try to use k-fold cross-validation to measure the performance of the model in prediction. In terms of reproducibility, we will list the references supporting the links in BN and clarify the node discretization schemes to make the methodology section more detailed and transparent.

Action: To improve the reproducibility of this study, we have added descriptions of methodology related details. Reproducibility is substantially improved because data is publicly available (Migliavacca and Musavi, 2021) and methods are more transparent and detailed in the revised manuscript. We list possible mechanisms of causal links in the BNs and relevant supporting references (articles on the relationships between climate, plant trait, and ecosystem functions). For the node discretization, we used an equal quantile [0,33.33%, 66.67%,100% percentile] discretization method, which resulted in a uniform discretization scheme for each node. To enhance the validation of BN, we also performed a 10-fold cross-validation by selecting nodes such as ETmax, GPPsat, and NEPmax for accuracy evaluation (comparing the agreement between the node statuses inferred from BN and the actual statuses).

**2.     Specific comments**

-     The data used in this study relies on a database collected by Magliavacca et al. (2021). In the cited paper, there are three complementary variables to quantify water use efficiency (G1, WUEt and uWUE); Rb was calculated in terms of both mean and max (95th percentile) values; and aCUE data is also available. The authors do not specify the criteria they followed when choosing which variables to include in the BN among the available variables in Magliavacca et al. (2021). Why was uWUE used and not the other water use efficiency metrics? How sensitive are the results to this choice? Why was the calculation of Rb unique in using the mean – compared to the other ecosystem function variables? If the network was based on expert knowledge, why to exclude aCUE which is explicitly included in the expert framework (Fig 1)?   What would be the effect of adding CUE as an extra constrain? How sensitive is the network to its inclusion?

Response: Thank you for your insightful comments. Since the scale of this BN network is relatively coarse, it is difficult to include various variables that represent similar meanings together. wUEt and uWUE are similar, although not identical, thus we have chosen only one of them. Similarly, using the 95th percentile of Rb rather than the mean of Rb may represent a different issue and meaning. In the modified version, CUE could be considered for inclusion in the BN, and CUE, as the terminal variable, should not have much impact on the sensitivity of the other variables.

Action: We have added the aCUE node to the BN structure and set aCUE to be causally controlled by GPPsat and Rb) (see Figure 1).

In addition, we described the possible influence of the choice of node variables and the actual meaning represented by the node variables on the results of BN in the Discussion section: "In addition to the causal relationship between nodes, the meaning represented by each node, the data source/ approach, and the spatial and temporal resolution may also have impacts on the results. For example, in this study, for multiple water use efficiency-related variables in the ref. (Migliavacca et al., 2021), we chose uWUE and for Rb we chose the mean value of Rb. The results of BN-based analysis may vary if different representations or meanings of ndes are selected. The way the data of each variable is observed/ produced, the spatial and temporal resolution of the data, etc. can also affect the understanding of the role of these variables in the data-driven BN. Some variables may be very important in the attribution of actual ecosystem function variation, but their importance may be underestimated due to limitations in the inherent observational accuracy of their data, and difference in their spatio-temporal scales from other variables. Thus, the alternative use of multiple derivatives of a variable and data generated by different methods for the construction of different BNs can help us to recognize how the uncertainty in the nodes and data can influence BN-based attribution findings."

- In the BN (Figure 1), the causal relationships plant traits – ecosystem functions were assigned based on expert knowledge (Reichstein et al, 2014). Then, the climatic variables and the respective causal relationships were added: how were the links climate-plant traits and climate-ecosystem functions determined? Considering that BN based on expert knowledge rely heavily on the prior understanding of the processes, the approach used to assign these links should be clearly stated in the methods.

Response: Thank you for your insightful comments. We have realised that the description of this section is not detailed. Therefore, in the revised version, we will list the literature supporting the possible impacts of climate variables and describe in more detail the possible impacts of climate variables.

Action: We have added an explanation of the possible mechanisms of the added causality and the related references in Table 2.

Table 2. Explanation of the added causal links between climate variable nodes, plant trait nodes, and ecosystem function variable nodes in the BNs.

| Casual links | | Explanation | References |
|---|---|---|---|
| Parent node | Child node | | |
| VPD | uWUE | uWUE= GPP· $VPD^{0.5}$/ET | (Zhou et al., 2014) |
| VPD | GSmax | stomatal and surface conductance declines under an increase in VPD | (Grossiord et al., 2020; Wever et al., 2002) |
| VPD | GPPsat | leaf and canopy photosynthetic rates decline when atmospheric VPD increases due to stomatal closure | (Yuan et al., 2019; Konings et al., 2017) |
| Tair | VPD | higher air temperature corresponds to higher saturated water vapor pressure and can drive an increase in VPD | (Yuan et al., 2019) |
| Tair | Hc | the temperature limitation on canopy height variation | (Moles et al., 2009) |

| | | | |
|---|---|---|---|
| Tair | Nmass | increase in air temperature may decrease plant nitrogen concentration and leaf nitrogen content. | (Weih and Karlsson, 2001; Reich and Oleksyn, 2004) |
| Tair | Rb | temperature strongly influences Rb through the laws of thermodynamics | (Davidson and Janssens, 2006; Enquist et al., 2003; Brown et al., 2004) |
| SWin | LAImax | solar radiation affects vegetation conditions and phenology | (Günter et al., 2008; Liu et al., 2016; Borchert et al., 2015; Wagner et al., 2017) |
| SWin | Hc | solar radiation affects the distribution and composition of ecosystems through photosynthesis and the water cycle | (Borchert et al., 2015; Guisan and Zimmermann, 2000; Piedallu and Gégout, 2007) |
| SWin | GPPsat | solar radiation affects ecosystem productivity and plant growth | (Monteith, 1972; Borchert et al., 2015; Guisan and Zimmermann, 2000) |
| P | Hc | the hydraulic limitation hypothesis on canopy height variation | (Moles et al., 2009; Ryan and Yoder, 1997; Koch et al., 2004) |
| P | Nmass | leaf nitrogen concentration per unit mass may decrease with increasing precipitation | (Santiago and Mulkey, 2005; Wright and Westoby, 2002) |
| CSWI | LAImax | soil moisture affects vegetation conditions | (Patanè, 2011) |
| CSWI | Rb | soil moisture affects the temperature dependence of ecosystem respiration | (Xu et al., 2004; Flanagan and Johnson, 2005; Wen et al., 2006) |
| CSWI | GPPsat | soil moisture can reduce GPP through ecosystem water stress | (Green et al., 2019) |

- Each plant-trait and ecosystem-function variable used in this study has a clear equivalent in the expert knowledge frame (upper panel in Figure 1) (Reichstein et al., 2014), except for AGB – which is unique in the authors model. Please provide more information about the assignment of links to this variable. How did you link Gsmax with AGB? Why not to link LAImax to AGB? AGB is confounded with wood density, height and other size metrics as plant diameter. Which variable on Reichstein's frame is being represented with AGB? How was the confounding controlled for? Can the authors ensure no circularity was added to the framework? This is because AGB from Globbiomass is inferred from models and algorithms that use as input remote indicator variables that are correlated with many of the other variables in the authors network. Finally, AGB from Globbiomass is subject to large error – how was this error controlled for?

Response: In the Globbiomass dataset (Santoro et al., 2018), the growing stock volume (GSV) estimates were obtained from spaceborne SAR (ALOS PALSAR, Envisat ASAR), optical (Landsat-7), LiDAR (ICESAT) and auxiliary datasets with multiple estimation procedures. AGB was obtained from GSV with a set of Biomass Expansion and Conversion Factors (BCEF) following approaches to extend on ground estimates of wood density and stem-to-total biomass expansion factors to obtain a global raster dataset. Therefore AGB can include more relevant information about Wood density etc. than LAImax. We will explain in more detail in the text the reasons for using AGB here to avoid confusion as you have suggested. Considering that the AGB does not only affect the LAImax through Hc, a link between the AGB and the LAI could be added as you have pointed out. In addition, the AGB data may indeed be subject to large errors and correlate with other variables in BN in this paper. We will add in the discussion section that there are possible implications of uncertainty in the data for nodes such as AGB.

*Reference*

*Santoro, M., Cartus, O., Mermoz, S., Bouvet, A., Le Toan, T., Carvalhais, N., Rozendaal, D., Herold, M., Avitabile, V., Quegan, S., Carreiras, J., Rauste, Y., Balzter, H., Schmullius, C., and Seifert, F. M.: A detailed portrait of the forest aboveground biomass pool for the year 2010 obtained from multiple remote sensing observations, 18932, 2018.*

Action: Based on your suggestion regarding the uncertainty of AGB and the associated potentially inappropriate causal links, after careful consideration, we removed the AGB node in the modified BN in the revised manuscript. The plant trait variables retained include LAImax, Hc, and Nmass, and the previous causality affecting AGB was modified to affect either LAImax or Hc.

-      More information is required regarding the specific criteria taken into account when defining the discretization thresholds. It is mentioned in the text (Lines 113-114) that the "meanings of the thresholds" were considered, but it is not completely clear how the thresholds were chosen nor what their meanings are. Ideally, BN models developed using different discretization methods should be considered and compared. If the results of such models are different, the choice of one method over the other should be justifiable (see e.g. Nojavan et al., 2017 - Comparative analysis of discretization methods in Bayesian networks).

Response: Thank you for your insightful comments. We will specify the basis for the discretization of each node (considering the distribution of values or the specific meaning of the thresholds). In addition, the use of different discretization schemes does have implications for causality and sensitivity analysis, which we will explain in more detail in the Discussion section.

Action: According to your comments, we used the equal quantile [0,33.33%, 66.67%, 100%] discretization method (Table 1), which makes the discretization of each node in a uniform way. Given the small amount of data available, a more detailed node discretization may make the compilation of BN difficult (due to the higher complexity of the conditional probability tables to be quantified). Therefore, for each node, we set only three levels (0-33.33%, 33.33%-66.67% and 66.67%-100%).

In the discussion section, we added: 'Different node discretization schemes may also affect the conditional probability table between nodes as well as the sensitivity (Nojavan A. et al., 2017). Other alternative discretization schemes with the commonly used three levels may also be effective, such as using 'mean-std' (mean minus 1 standard deviation) and 'mean+std' (mean plus 1 standard deviation) as discretization thresholds, which will result in a change in the relationship between BN nodes. And further if extreme values such as 5th and 95th pencentile are used in the node value discretization, it may be beneficial on quantifying the causal control of extreme conditions of nodes on other nodes.'

-      Not all FLUXNET stations used in Magliavacca et al. (2021) have data available for all the variables. Only 94 out of 203 sites have data regarding vegetation structure, N%, LAImax, Hc and AGB. This means that there is a substantial amount of missing data in the model. Please report the missing fractions in the manuscript, or another indicator of the amount of missing data treated with the Expectation-Maximization method that is mentioned in the text (Line 131). Also, how robust are the results to the imputation methods used? It is critical to show that the results are not dependent on data imputation when a large amount of data is missing.

Response: Thank you for your insightful comments. We will explain this issue in more detail in the text. The use of Expectation-maximization has the potential to introduce uncertainty and bias the relationship between other nodes and plant trait nodes in favour of the 94 sites with available plant trait data. The possible effects of this data incompleteness and the resulting uncertainty will be described in more detail in the Discussion section.

Action: We have described the data incompleteness in the revised manuscript: 'Of the total 202 sites (Migliavacca and Musavi, 2021), 101 sites have Nmass data, 153 sites have LAImax data, and 199 sites have Hc data. Only 98 have data on all these three plant trait variables.'

We also mentioned the possible influence of using the EM algorithm under data incompleteness conditions: 'We compiled two different BNs (i.e., BN_plant_trait and BN_plant_trait_climate) (Figure 3) and found that the probability distributions of the values of the common nodes (ecosystem function and plant trait variable nodes) differed a little (e.g., in the probability distribution of LAImax, Hc, and Nmass) between the two BNs. Compared to BN_plant_trait, in BN_plant_trait_climate, the climate variables of sites with missing plant trait data forced the changes in the probability distributions of LAImax, Hc, and Nmass. In the EM algorithm, for sites with missing plant trait data, existing relationships (obtained from observations from other sites) between plant trait variables and climate variables are used in the data interpolation of plant trait variables. In BN_plant_trait_climate, the added linkages of climate variables to plant trait variables resulted in higher probability values of the low-value status of the plant trait variables.'

- One core question arising from this study is: how to show that the artificial-rules-based model can reveal the real rules compared with a data-driven model? It is, therefore, necessary to show validation results in the paper. What validation methods did the authors use? how does this validation result compare to typical standard seen in similar studies? If a validation (i.e. k-fold), and robustness checks are done and reported, then the model results can be interpreted with more certainty. However, with the available data in the current version of the manuscript, the model performance cannot be assessed. We believe, that even though the BN was calculated for categorical data instead of continuous data, there is always a need to show that the model predicted well through model validation techniques. Validation results are also important for comparison/verification with future ecological-knowledge-based models.

Response: Thank you for your insightful comments. Despite the limited amount of data, we will consider doing a k-fold cross-validation. The level of the highest probability of a node predicted by BN can be compared and validated against the actual values (e.g. reporting error matrix).

Action: We added a 10-fold cross-validation to evaluate the performance of BN_plant_trait_climate, leaving about 20 sites per fold for validation. Classification accuracy was evaluated for ETmax, GPPsat, and NEPmax (the statuses of these nodes inferred from the BN were compared with the actual statuses and the classification accuracy was calculated). In general, the classification accuracy was good (Table S1). The little a bit low accuracy of ETmax may be due to the fact that only GSmax was set in BN to directly affect ETmax, which may not be sufficient to fully explain the variation of ETmax. In addition, the wide range of climatic and biome heterogeneity may also affect the accuracy of cross-validation, especially when very outlier sites (representing particular climatic and biome types) are present in the validation set.

Table S1. The 10-fold cross-validation of BN_plant_trait_climate on ETmax, NEPmax, and GPPsat nodes.

| Validation node | Actual status | Predicted status | | | Accuracy |
|---|---|---|---|---|---|
| | | low | medium | high | |
| ETmax | low | 47 | 16 | 5 | 60.9% |
| | medium | 12 | 35 | 14 | |
| | high | 9 | 21 | 41 | |
| NEPmax | low | 62 | 5 | 0 | 84.2% |
| | medium | 5 | 52 | 11 | |
| | high | 0 | 11 | 56 | |

| GPPsat | low | 62 | 5 | 0 | 75.2% |
|--------|--------|----|----|----|-------|
| | medium | 6 | 45 | 17 | |
| | high | 0 | 22 | 45 | |

Note: Low, medium, and high status of ETmax correspond respectively to 0.059 to 0.17, 0.17 to 0.23, and 0.23 to 0.423 (Table 1 in the main text). Low, medium, and high statuses of NEPmax correspond respectively to 1.953 to 15.3, 15.3 to 24.4, and 24.4 to 42.82 (Table 1). Low, medium, and high status of GPPsat correspond respectively to 3.042 to 17.49, 17.49 to 27.74, and 27.74 to 47.6 (Table 1).

- Finally, to prevent confusion and over interpretation of the result, the authors should acknowledge that BN are not necessarily causal networks, they are essentially a set of conditional (in)dependencies that factorize the joint probability distribution of all the variables. Causal deductions hence may not be made (Ramazi et al., 2020 - Exploiting the full potential of Bayesian networks in predictive ecology).

Response: Thank you for your insightful comments. BN relies to some extent on the links between nodes set by the modeller. We will mention this limitation of BN in the discussion section.

Action: The word 'causal networks' is replaced with 'causal graphical models' in the revised manuscript.

In the discussion section, we also list the potential limitations of BN in terms of causal representation: " BN essentially factorizes the joint probability distribution among data variables into a series of conditional probability distributions (Ramazi et al., 2021), and the reliability of this approach relies on the setting of causal control relationships among nodes. Expert knowledge is thus critical in the construction of BNs, especially when modeling complex systems. "

**3. Technical corrections**

**3.1. Introduction**

- Since the main focus of this study are ecosystem functions, a clear and concise definition of this term is needed in the Introduction section. Also, it would be useful adding some supporting references regarding the theory linking the functional traits included in the paper and Reichstein et al.'s frame.

Response: Thank you for your insightful comments. We will add more relevant texts and references in Introduction section.

Action: elaborated in the Introduction section: 'Ecosystem function is the capacity of natural processes and components to provide goods and services that satisfy human needs, either directly or indirectly (de Groot et al., 2002). Ecosystem functions include the physicochemical and biological processes within the ecosystem to maintain terrestrial life. Terrestrial ecosystems have provide a variety of important ecosystem functions for our society (Manning et al., 2018). Plant traits' role as important determinants of ecosystem functions has been widely recognized (Chapin Iii et al., 2000), various trait syndromes can result in distinct broad differences in ecosystem functions (Reichstein et al., 2014). In the context of global climate change, it is also essential to understand the potential changes in ecosystem functions (Grimm et al., 2013).'

- There are some terms used along the paper referring to climate variables, vegetation structure variables, or ecosystem functions. These terms change along the manuscript. Try to use a consistent terminology. Examples of these terms are:

Line 27: "complex ecosystems", "environmental systems".

Line 37: "environmental conditions"

Line 60: "environments"

Line 158: "ecosystem service functions"

Line 271: "ecosystem systems"

Response: Thank you for your insightful comments. These will be revised and unified.

Action: Unified.

- Lines 66-67: The paper by Gregorutti et al. (2017) is used to support the statement that IMP-based attribution can be unreliable when the aim is explaining systematic causality. The cited paper does not discuss systematic causality. Please support this idea with appropriate references.

Response: Thank you for your insightful comments. IMP-based attribution has difficulty in dealing with mutual non-independence in predictors. For example, if both NDVI and EVI are used in model A and only NDVI is used in model B, then the IMP of NDVI in model A will be lower than that in model B. We will use a more appropriate literature or modify the description here.

Action: revised as: 'However, IMP-based attribution to the target variable can also be unreliable if considerable confounders and correlations between predictor variables exist (Strobl et al., 2008; Toloşi and Lengauer, 2011). The less relevant predictors can replace the predictive predictors (due to correlation) and thus receive undeserved high feature importance (Strobl et al., 2008). Correlations between predictors can lead to biased feature-importance-based findings. It is thus important to recognize the difference between correlation and causality in these approaches, emphasize detailed causal relations between features, rather than the unreliable feature importance rankings generated from correlated features.'

**3.2. Methodology**

- Line 95: "Climatic variables:…"

Response: It will be added.

Action: It is added.

- Table 1: Though the detailed methods for the ecosystem functions' calculation is in Magliavacca et al. (2021), provide a complete summary for each variable in the column "Approach". E.g. if the percentile used in the calculations will be reported, report it for all the variables consistently (GSmax – 90th percentile). The use of medians instead of means for some of the variables may also be important information.

Response: Thank you for your suggestion, we will add this column.

Action: The 'Approach' column is more detailed based on your suggestion (see Table 1).

- Line 119: Figure 1 is presented as a column with three panels. It is not clear what the author refers to when pointing to the lower left panel.

Response: The figure caption will be corrected.

Action: Figure caption is corrected.

- Lines 142-147: This sentence is quite long. Try to split it or make it shorter.

Response: It will be revised.

Action: modified into shorter sentences: 'Further, to clarify the adding-values of considering causality in the attribution analysis of controls on ecosystem functions, the results of the BN-based sensitivity analysis (BN_sens) were compared with other two approaches. They are the results of the absolute values of additional linear correlation analysis (linear_corr) in this study and the findings from the ref. (Migliavacca et al., 2021) using RF feature importance (RF_imp).'

**3.3.    Results**

- Line 154: Incomplete sentence at the end of this line: "… SWin, VPD, and showed…"

Response: It will be revised.

Action: revised as 'SWin and VPD showed negative correlations with these ecosystem function variables.'

- Figure 3: In the text, some information is extracted from this figure regarding the correlations climate vs. ecosystem functions and ecosystem functions vs. ecosystem functions. More information can be extracted from this figure regarding the correlations ecosystem function vs. plat traits and plat traits vs. climate.

Response: We will add some description text with this information.

Action: added: 'LAImax/ Hc showed significant positive relationships with most of the ecosystem function variables and significant negative relationships with SWin and VPD. Nmass only showed a positive relationship with ETmax.'

- Figure 4: Can the display of this figure be improved? E.g. locating the tables in a more equidistant layout.

Response: It will be improved.

Action: The figure is Improved.

- Lines 195-196: When looking at Figure 1, it is not clear what the author is referring to with the loop of Tair controlling LAImax. What variables are included in this loop? This word should be used with caution since it could be taken as equivalent to a feedback.

Response: We will check and update the structure of BN and re-analyse the relevant results.

Action: This sentence is deleted. Because we changed the structure of BN, the results such as sensitivity analysis are different from those in the previous version. We have analyzed the new results. We added: 'For example, in BN_plant_trait_climate, for GPPsat, a decrease in the sensitivity of GPPsat to LAImax and an increase in the sensitivity to Tair was observed after the causal chain of Tair influencing Hc, LAImax, and then GPPsat was set. This can be explained by the fact that higher temperatures promote vegetation growth and thus may increase LAImax, which then indirectly alters the probability distribution of the GPPsat node. In previous studies based on statistical methods that did not consider the chain causality, this indirect control on GPPsat from Tair may have been included in the contribution of LAImax to GPPsat. Similarly, a chain causality of P by first affecting Nmass and

then indirectly GPPsat was also found. However, the effect of P by first affecting Hc, LAImax, and then indirectly affecting ETmax and GSmax appears to be not large.'

**3.4.    Discussion**

-    The interactions among plant traits, climate and ecosystem function variables may be complex when high-order effects are considered. Does the author think these effects play an important role when trying to explain the causal links to ecosystem functions? If these effects are not considered in this study, this limitation should be stated in the Discussion section.

Response: Thank you for your insightful comments. Our BN does not take into account the higher order effects of each node. This is of interest, but may have higher requirements for expert knowledge. It may be useful to combine this with the relevant literature that studied the impacts of higher order effects. In the discussion section we refer to the potential of using BNs to analyse the effects between such inclusion of higher order effects.

Action: Added in the discussion section: 'When considering higher-order effects (Bairey et al., 2016), the relationships between plant trait, climate variables, and ecosystem function variables can be very complex. One variable may affect the relationship between two other variables rather than directly affecting these two variables (Bairey et al., 2016). BN may have limitations in directly analyzing such higher-order effects because BN requires the modeler to explicitly set direct causal relationships between nodes. To analyze the higher-order effects, we can add nodes that directly represent the relationship between the variables. For example, the correlation coefficient of two variables can be used as a node and this node is connected to other nodes in the BN so that the control effect of other nodes on this correlation coefficient can be explored. Such implements may be useful to deepen the impact of various higher order effects.'

**Response to Referee #2**

**1. General comments**

The study describes exemplarily the construction of a network linking plant traits and climatic drivers not only with a statistical background but by taking into account causal linkages. Using a Bayesian Network (BN), expert knowledge is introduced to evaluate the causal effects of climate variables for ecosystem functions. The main achievement and argument is that this type of analysis goes beyond usual statistical relationships which often fail to reveal indirect effects and trade-offs. Although this approach is appealing and from an ecological point of view very promising, the manuscript does not provide a proper validation of the method. The increasing availability of data such as collected within the FLUXNET community hopefully will further trigger new ways of exploring the connection of environmental conditions with the evolving plant community and at the same time allow to test and consolidate analysis tools. Here, the method would benefit from better methodological clarification, description of data use, validation and presentation of the results which are detailed below. The paper needs major revisions before publication.

Response: Thank you for your insightful comments, which have been very helpful in improving the manuscript. This manuscript will be revised in accordance with your comments. In terms of model validation, we will try to use k-fold cross-validation to measure the performance of the model in prediction. In terms of reproducibility, we will list the references supporting the links in BN and clarify the node discretization schemes to make the methodology section more detailed and transparent.

Action: To improve the reproducibility of this study, we have added descriptions of methodology related details. Reproducibility is substantially improved because data is publicly available (Migliavacca and Musavi, 2021) and methods are more transparent and detailed in the revised manuscript. We list possible mechanisms of causal links in the BNs and relevant supporting references (articles on the relationships between climate, plant trait, and ecosystem functions). To enhance the validation of BN, we also performed a 10-fold cross-validation by selecting nodes such as ETmax, GPPsat, and NEPmax for accuracy evaluation (comparing the agreement between the node statuses inferred from BN and the actual statuses).

**2. Specific comments**

- The text includes various repetitions when stressing that the new method is superior to usual analyses. Please be more concise when making this point (e.g. in introduction and discussion) or more specific when certain aspects are described in detail (e.g. in results). The re-occurring statement is not strengthening the argument.

Response: Thank you for your insightful comments. We will improve the relevant descriptions.

Action: The relevant description has been more concise in the revised version.

- Although the data base for the BN is given in table 1 in detail, the choice of the variables does not become clear. Which variables were taken into account and why? Some variables are taken as is and some averaged. Please state as well the temporal resolutions of original variables and averages (why mean values and not medians?). Also the choice of the intervals for discretization (right column in table 1) is not motivated – please provide more detail and reasoning.

Response: Thank you for your insightful comments. We will describe in more detail the choice of variables and what these variables specifically represent, etc. We will also specify the basis for the discretization of each node (considering the distribution of values or the specific meaning of the thresholds). In addition, the use of different discretization schemes does have implications for causality and sensitivity analysis, which we will explain in more detail in the Discussion section.

Action: We used the data from the ref. (Migliavacca et al., 2021) directly without additional processing. Some variables in the dataset are the $90^{th}$/ $95^{th}$ percentile values extracted from the daily or

half-hourly values, while others are the mean/ median. We have added relevant information and also the data approach containing temporal resolutions of original data in the 'Approach' column in Table 1.

Table 1. The variables used and the discretization of their values in BN.

| Variable node | Definition and units | Type | Approach (Migliavacca et al., 2021) | Discretization in BN (equal quantile thresholds: 0%, 33.33%, 66.67%, and 100% percentile values) |
|---|---|---|---|---|
| uWUE | underlying Water Use Efficiency [gC kPa^0.5 kgH$_2$O$^{-1}$] | Ecosystem function | It was calculated from GPP, VPD, and ET (Zhou et al., 2014). The median of the half-hourly retained uWUE values was used for each site. It was further filtered by the following conditions: (i) SWin > 200 W m$^{-2}$; (ii) no precipitation event for the last 24 hours, when precipitation data are available; and (iii) during the growing season: daily GPP > 30% of its seasonal amplitude. | 0.068, 2.51, 3.18, 5.332 |
| ETmax | maximum evapotranspiration in the growing season [mm] | Ecosystem function | ETmax was computed as the 95th percentile of ET in the growing season. It was also filtered by the same filtering applied for the uWUE calculation. | 0.059, 0.17, 0.23, 0.423 |
| GSmax | maximum surface conductance [m s$^{-1}$] | Ecosystem function | GSmax was computed by inverting the Penman-Monteith equation after calculating the aerodynamic conductance. The 90th percentile of the half-hourly GS of each site was calculated and used as the GSmax of each site. | 0.0013, 0.0077, 0.0123, 0.0566 |
| NEPmax | maximum net CO2 uptake of the ecosystem [umol CO$_2$ m$^{-2}$ s$^{-1}$] | Ecosystem function | NEPmax was computed as the 90th percentile of the half-hourly net ecosystem production in the growing season (when daily GPP is > 30% of the GPP amplitude). | 1.953, 15.3, 24.4, 42.82 |
| GPPsat | Gross Primary Productivity at light saturation [umol CO$_2$ m$^{-2}$ s$^{-1}$] | Ecosystem function | GPPsat was computed as the 90th percentile estimated from half-hourly data by fitting the hyperbolic light response curves. The 90th percentile | 3.042, 17.49, 27.74, 47.6 |

| | | | from the GPPsat estimates of each site was extracted. | |
|---|---|---|---|---|
| Rb | Mean basal ecosystem respiration at a reference temperature of 15 °C [umol $CO_2$ $m^{-2}$ $s^{-1}$] | Ecosystem function | Rb was derived from night-time NEE measurements. For each site, the mean of the daily Rb value were computed. | 0.144, 2.07, 3.12, 10.67 |
| aCUE | apparent carbon-use efficiency | Ecosystem function | aCUE was calculated by aCUE = 1-(Rb/GPP) and the median value of daily aCUE is used. | -1.19, 0.4, 0.74, 1 |
| Nmass | ecosystem scale foliar nitrogen concentration [gN $100$ $g^{-1}$] | Plant trait | Nmass was computed as the community-weighted average of foliar N% of the major species at the site sampled at the peak of the growing season or gathered from the literature (Musavi et al., 2016, 2015; Fleischer et al., 2015; Flechard et al., 2020). | 0.65, 1.15, 1.76, 4.44 |
| LAImax | Maximum Leaf Area Index [$m^2$ $m^{-2}$] | Plant trait | LAImax was collected from the literature (Migliavacca et al., 2011; Flechard et al., 2020), the FLUXNET Biological Ancillary Data Management (BADM) product, and/or site principal investigators. | 0.17, 2.27, 4.5, 12.9 |
| Hc | Maximum vegetation height [m] | Plant trait | Hc was collected from the literature (Migliavacca et al., 2011; Flechard et al., 2020), the BADM product, and/or site principal investigators. | 0.04, 1.7, 16.0, 80.1 |
| SWin | Mean incoming shortwave radiation [W $m^{-2}$] | Climate | SWin was from FLUXNET data. | 54.43, 134.18, 182.44, 266.04 |
| Tair | Mean temperature [degree C] | Climate | Tair was from FLUXNET data. | -10.45, 6.62, 14.73, 28.1 |
| VPD | Mean Vapor Pressure Deficit [hPa] | Climate | VPD was from FLUXNET data. | 0.62, 3.38, 5.76, 26.08 |
| P | Mean annual precipitation [cm/year] | Climate | P was from FLUXNET data. | 5.51, 45.28, 79.29, 256.61 |
| CSWI | cumulative soil water index | Climate-related soil | CSWI was computed as a measure of water availability (Nelson et al., 2018). | -93.49, -1.24, 2.01, 4.47 |

| | | water availability | | |
| --- | --- | --- | --- | --- |

We described the possible influence of the choice of node variables and the actual meaning represented by the node variables on the results of BN in the Discussion section: "In addition to the causal relationship between nodes, the meaning represented by each node, the data source/ approach, and the spatial and temporal resolution may also have impacts on the results. For example, in this study, for multiple water use efficiency-related variables in the ref. (Migliavacca et al., 2021), we chosen uWUE and for Rb we chosen the mean value of Rb. The results of BN-based analysis may vary if different representations or meanings of ndes are selected. The way the data of each variable is observed/ produced, the spatial and temporal resolution of the data, etc. can also affect the understanding of the role of these variables in the data-driven BN. Some variables may be very important in the attribution of actual ecosystem function variation, but their importance may be underestimated due to limitations in the inherent observational accuracy of their data, and difference in their spatio-temporal scales from other variables. Thus, the alternative use of multiple derivatives of a variable and data generated by different methods for the construction of different BNs can help us to recognize how the uncertainty in the nodes and data can influence BN-based attribution findings."

In the revised manuscript, we used the equal quantile [0,33.33%, 66.67%, 100%] discretization method (Table 1), which makes the discretization of each node in a uniform way. Given the small amount of data available, a more detailed node discretization may make the compilation of BN difficult (due to the higher complexity of the conditional probability tables to be quantified). Therefore, for each node, we set only three levels (0-33.33%, 33.33%-66.67% and 66.67%-100%).

In the discussion section, we added: 'Different node discretization schemes may also affect the conditional probability table between nodes as well as the sensitivity (Nojavan A. et al., 2017). Other alternative discretization schemes with the commonly used three levels may also be effective, such as using 'mean-std' (mean minus 1 standard deviation) and 'mean+std' (mean plus 1 standard deviation) as discretization thresholds, which will result in a change in the relationship between BN nodes. And further if extreme values such as 5th and 95th pencentile are used in the node value discretization, it may be beneficial on quantifying the causal control of extreme conditions of nodes on other nodes.'

- The interesting part of constructing the BN in section 2.2 is not transparent. On which basis is the expert knowledge extracted from Reichstein et al. (2014) and how is it transferred to the BN? When the authors main agenda is to promote their new analysis method, it would be good to give more insights in the process of finding the linkages that should be considered.

Response: Thank you for your insightful comments. We will add literature that has a description of how to transfer expert knowledge to BN, as well as adding more specific information on the impact of climate variables. We will make the methodology section more detailed and transparent.

Action: We have added an explanation of the possible mechanisms of the added causality and the related references in Table 2.

Table 2. Explanation of the added causal links between climate variable nodes, plant trait nodes, and ecosystem function variable nodes in the BNs.

| Casual links | | Explanation | References |
| --- | --- | --- | --- |
| Parent node | Child node | | |
| VPD | uWUE | uWUE= GPP· $VPD^{0.5}$/ET | (Zhou et al., 2014) |

| | | | |
|---|---|---|---|
| VPD | GSmax | stomatal and surface conductance declines under an increase in VPD | (Grossiord et al., 2020; Wever et al., 2002) |
| VPD | GPPsat | leaf and canopy photosynthetic rates decline when atmospheric VPD increases due to stomatal closure | (Yuan et al., 2019; Konings et al., 2017) |
| Tair | VPD | higher air temperature corresponds to higher saturated water vapor pressure and can drive an increase in VPD | (Yuan et al., 2019) |
| Tair | Hc | the temperature limitation on canopy height variation | (Moles et al., 2009) |
| Tair | Nmass | increase in air temperature may decrease plant nitrogen concentration and leaf nitrogen content. | (Weih and Karlsson, 2001; Reich and Oleksyn, 2004) |
| Tair | Rb | temperature strongly influences Rb through the laws of thermodynamics | (Davidson and Janssens, 2006; Enquist et al., 2003; Brown et al., 2004) |
| SWin | LAImax | solar radiation affects vegetation conditions and phenology | (Günter et al., 2008; Liu et al., 2016; Borchert et al., 2015; Wagner et al., 2017) |
| SWin | Hc | solar radiation affects the distribution and composition of ecosystems through photosynthesis and the water cycle | (Borchert et al., 2015; Guisan and Zimmermann, 2000; Piedallu and Gégout, 2007) |
| SWin | GPPsat | solar radiation affects ecosystem productivity and plant growth | (Monteith, 1972; Borchert et al., 2015; Guisan and Zimmermann, 2000) |
| P | Hc | the hydraulic limitation hypothesis on canopy height variation | (Moles et al., 2009; Ryan and Yoder, 1997; Koch et al., 2004) |
| P | Nmass | leaf nitrogen concentration per unit mass may decrease with increasing precipitation | (Santiago and Mulkey, 2005; Wright and Westoby, 2002) |
| CSWI | LAImax | soil moisture affects vegetation conditions | (Patanè, 2011) |
| CSWI | Rb | soil moisture affects the temperature dependence of ecosystem respiration | (Xu et al., 2004; Flanagan and Johnson, 2005; Wen et al., 2006) |
| CSWI | GPPsat | soil moisture can reduce GPP through ecosystem water stress | (Green et al., 2019) |

- The result section would benefit from a better description of the results of both methods. Reducing the text with general statements should give enough space for guiding through figure 4 and highlighting the benefits of the second approach. How do you motivate this statement when e.g. comparing the results for AGB in the BN-plant-trait-climate in comparison to the BN-plant-trait?

Response: Thank you for your insightful comments. We will add more analytical and explanatory descriptions related to Figure 4, comparing the differences between the two BNs.

Action: We compared the differences of the two compiled BNs in Figure 3: 'We compiled two different BNs (i.e., BN_plant_trait and BN_plant_trait_climate) (Figure 3) and found that the probability distributions of the values of the common nodes (ecosystem function and plant trait variable nodes) differed a little (e.g., in the probability distribution of LAImax, Hc, and Nmass) between the two BNs. Compared to BN_plant_trait, in BN_plant_trait_climate, the climate variables of sites with missing plant trait data forced the changes in the probability distributions of LAImax, Hc, and Nmass. In the EM algorithm, for sites with missing plant trait data, existing relationships (obtained from observations from other sites) between plant trait variables and climate variables are used in the data interpolation of plant trait variables. In BN_plant_trait_climate, the added linkages of climate variables to plant trait variables resulted in higher probability values of the low-value status of the plant trait variables.'

- One major concern is a validation. A presentation of a data-driven method without a validation can hardly be recommended for publication. Please not only provide one but also make clear which data are used for building the model, getting the results and performing the validation.

Response: Thank you for your insightful comments. Despite the limited amount of data, we will consider doing a k-fold cross-validation. The level of the highest probability of a node predicted by BN can be compared and validated against the actual values (e.g. reporting error matrix).

Action: We added a 10-fold cross-validation to evaluate the performance of BN_plant_trait_climate, leaving about 20 sites per fold for validation. Classification accuracy was evaluated for ETmax, GPPsat, and NEPmax (the statuses of these nodes inferred from the BN were compared with the actual statuses and the classification accuracy was calculated). In general, the classification accuracy was good (Table S1). The low accuracy of ETmax may be due to the fact that only GSmax was set in BN to directly affect ETmax, which may not be sufficient to fully explain the variation of ETmax. In addition, the wide range of climatic and biome heterogeneity may also affect the accuracy of cross-validation, especially when very outlier sites (representing particular climatic and biome types) are present in the validation set.

Table S1. The 10-fold cross-validation of BN_plant_trait_climate on ETmax, NEPmax, and GPPsat nodes.

| Validation node | Actual status | Predicted status | | | Accuracy |
| --- | --- | --- | --- | --- | --- |
| | | low | medium | high | |
| ETmax | low | 47 | 16 | 5 | 60.9% |
| | medium | 12 | 35 | 14 | |
| | high | 9 | 21 | 41 | |
| NEPmax | low | 62 | 5 | 0 | 84.2% |
| | medium | 5 | 52 | 11 | |
| | high | 0 | 11 | 56 | |
| GPPsat | low | 62 | 5 | 0 | 75.2% |
| | medium | 6 | 45 | 17 | |
| | high | 0 | 22 | 45 | |

Note: Low, medium, and high status of ETmax correspond respectively to 0.059 to 0.17, 0.17 to 0.23, and 0.23 to 0.423 (Table 1 in the main text). Low, medium, and high statuses of NEPmax correspond respectively to 1.953 to 15.3, 15.3 to 24.4, and 24.4 to 42.82 (Table 1). Low, medium, and high status of GPPsat correspond respectively to 3.042 to 17.49, 17.49 to 27.74, and 27.74 to 47.6 (Table 1).

**3. Technical remarks**

L 20 and 31: The term 'emphasized' seems not appropriate in this context. Please be more specific what you mean here.

Response: It will be revised more specificly.

Action: revised as: 'Using statistical methods that **not directly representing the causality between variables** to attribute climate and plant traits to control ecosystem function may produce biased perceptions. We revisit this issue using a Bayesian network (BN) capable of quantifying causality **by conditional probability tables.**'

L 36: 'Changes in climate change' is misleading – please modify.

Response: It will be revised.

Action: revised as: 'The response of terrestrial ecosystem function to changes in climate, plant traits, and the corresponding mechanisms'

L 64: The sentence is very long and could be split into two.

Response: It will be revised.

Action: revised as: 'For machine learning techniques, although current common algorithms such as RF (Migliavacca et al., 2021) can report the importance of features (IMP) to measure their contributions to the prediction model. However, IMP-based attribution to the target variable can also be unreliable if considerable confounders and correlations between predictor variables exist (Strobl et al., 2008; Toloşi and Lengauer, 2011).'

L 67: Also very long sentence which makes me wonder, if you assume all relations in these systems to be causal, which they are of course not. Please clarify.

Response: It will be clarified.

Action: revised as 'Correlations between predictors can lead to biased feature-importance-based findings. It is thus important to recognize the difference between correlation and causality in these approaches, represent detailed causal relations between features, rather than the unreliable feature importance rankings generated from correlated features.'

L 96: Including the cumulative soil water index means that a variable is chosen which is already the result of precipitation and evapotranspiration. How do you deal with the interdependency of the variables?

Response: Here we follow the paradigm of a process-based or water balance approach, where ET is usually the output and precipitation is the forcing driver, and soil water comes from precipitation and contribute to ET with water supply.

Action: added in Discussion section: 'In addition, some variables such as soil moisture may be difficult to obtain due to the lack of continuous site-scale long-term observations. Using the water balance method to calculate CSWI as a proxy may introduce errors. Since the CSWI calculation method relies on P, etc., the obtained relationship between P, CSWI, and other nodes may have contained empirical components. If the availability of measurements of some nodes is low, modelers should be cautious about the empirical dependencies with other nodes that may be included in the alternative data approaches.'

Fig. 2: please explain the black dots in the figures.

Response: This is the position of each value on the horizontal axis. We will explain it.

Action: explained in figure caption: 'Dark green dots on the horizontal axis are the positions of variable values.'.

L 142: Another very long sentence on a complex issue. A stepwise approach would increase readibility.

Response: It will be revised.

Action: modified into shorter sentences: 'Further, to clarify the adding-values of considering causality in the attribution analysis of controls on ecosystem functions, the results of the BN-based sensitivity analysis (BN_sens) were compared with other two approaches. They are the results of the absolute values of additional linear correlation analysis (linear_corr) in this study and the findings from the ref. (Migliavacca et al., 2021) using RF feature importance (RF_imp).'

L 163: How do you evaluate the compilation as being 'successful'? Which criteria are fulfilled?

Response: We can use the k-fold cross-validation described above to further evaluate the compilation of BN.

Action: We performed a 10-fold cross-validation (described above) and reported the accuracy: '

The 10-fold cross-validation of the nodes ETmax, GPPsat, and NEPmax showed relatively high accuracy. The classification accuracy (Table S1) of the status of ETmax was 60.9%, the classification accuracy of the status of NEPmax was 84.2% and the classification accuracy of the status of GPPsat was 75.2%.'

Fig. 4: Values and text in the figure are very small. Why did you choose '?' as a separator between mean and standard deviation?

Response: We will make the text of this figure larger. '?' can be replaced by '±'.

Action: The figure resolution and quality are improved. The figure caption is also modified: 'At the bottom part of each node, the left and right side values of the '±' are the mean and standard deviation of the distribution, respectively.'

L 190: As an example for the wish for a better presentation of the results please give more reasoning for the statement that climate variables 'showed a role beyond plant traits'. Without an understandable link to the results shown, a sentence like this is appropriate in the discussion.

Response: We will discuss this issue in the discussion section.

Action: These sentences have been deleted.

L 224: The methods described in the caption and the text should be moved to the methods section! Here, please elaborate more on the explanation of the very valuable figure 6.

Response: We will place the method-related content from the caption in the methodology section. In addition, more analyses related to Fig. 6 can be mined and provided in the manuscript.

Action: The methods described in the caption and the text have been moved to the methods section. Figure 5 is updated due to BN structure change in the revised manuscript and the related analysis is also updated: 'All three methods show the importance of the plant trait variables in explaining the variation of various ecosystem function variables. LAImax was the most important of the three methods in explaining the variation of maximum ecosystem productivity properties (corresponding to PC1). In contrast to the results of the other two methods, in linear_corr, SWin and VPD were the least important, while P was more important. Comparing RF_imp and BN_sens, the overall pattern of importance is similar, but there are differences. For water-use strategies (corresponding to PC2), Hc is ranked first and LAI last in RF_imp, but in BN_sens, LAI is slightly more important than Hc. In linear_corr, Hc and LAI are of similar importance. For PC3, VPD ranks first and is more important than Tair in RF_imp. But in BN_sens, Tair is more important than VPD. Among the three moisture-related climate variables (i.e., VPD, P, and CSWI), CSWI appears to be the least important in RF_imp but is comparable to VPD in BN_sens.'

| | Methods | Nmass | LAImax | Hc | SWin | Tair | VPD | P | CSWI |
|---|---|---|---|---|---|---|---|---|---|
| PC1 | RF_imp | 10.80% | 16.60% | 14.50% | 7.60% | 9.10% | 11.70% | 6.70% | 4.00% |
| PC2 | RF_imp | 5.10% | 4.50% | 14.90% | 10.70% | 11.20% | 7.40% | 9.00% | 8.30% |
| PC3 | RF_imp | 7.00% | 2.80% | 5.40% | 9.30% | 8.00% | 15.40% | 6.50% | 4.90% |
| | | | | | | | | | |
| GPPsat | BN_sens | 0.0635 | 0.1980 | 0.0766 | 0.0299 | 0.0116 | 0.0221 | 0.0232 | 0.0380 |
| NEPmax | BN_sens | 0.0464 | 0.1482 | 0.0588 | 0.0168 | 0.0064 | 0.0065 | 0.0181 | 0.0142 |
| ETmax | BN_sens | 0.0006 | 0.0424 | 0.0076 | 0.0028 | 0.0063 | 0.0174 | 0.0006 | 0.0122 |
| uWUE | BN_sens | 0.0228 | 0.0321 | 0.0174 | 0.0012 | 0.0023 | 0.0080 | 0.0066 | 0.0072 |
| GSmax | BN_sens | 0.0022 | 0.1464 | 0.0246 | 0.0115 | 0.0239 | 0.0793 | 0.0019 | 0.0429 |
| Rb | BN_sens | 0.0880 | 0.0043 | 0.0021 | 0.0106 | 0.1177 | 0.0317 | 0.0053 | 0.0602 |
| aCUE | BN_sens | 0.0049 | 0.0138 | 0.0056 | 0.0033 | 0.0117 | 0.0009 | 0.0004 | 0.0007 |
| | | | | | | | | | |
| GPPsat | linear_corr | | 0.67 | 0.46 | 0.13 | | 0.20 | 0.48 | |
| NEPmax | linear_corr | | 0.63 | 0.56 | | | 0.13 | 0.48 | |
| ETmax | linear_corr | 0.44 | | | | | | 0.47 | 0.30 |
| uWUE | linear_corr | | 0.45 | 0.47 | 0.15 | | | | |
| GSmax | linear_corr | | | | | | 0.28 | | |
| Rb | linear_corr | | 0.57 | 0.35 | 0.21 | | 0.33 | 0.43 | |
| aCUE | linear_corr | | | | | | | | |

Figure 5. Comparisons of relationships of ecosystem functional variables to plant traits and climate variables in different analyses. Method RF_imp is Random forest variable importance (Migliavacca et al., 2021) (see Methodology section). Method linear_corr is Linear correlation analysis with the absolute values of Pearson correlation coefficients (see Methodology section). Method BN_sens is a BN-based sensitivity analysis with sensitivity values MI reported. The values in each method group are in red for high values and in blue for low values.

L 281: The idea of extending the causal linkages to the temporal dimension is intriguing but opens the problem of non-independent variables. Do you have an idea how to treat causally linked and dependent variables in this approach?

Response: Thank you for your insightful comments. Completely resolving the independence of the relationship between the temporal dimensions of these variables may be controversial. On the one hand, this independence may depend on the time scale of the study, and on the other hand, it may require us to add the control of the precipitation constraint (i.e., we can constrain our study aim to analyse the ET-VPD-CSWI relationship in time periods split by precipitation event because a precipitation event is enough to interrupt the extension of this relationship). We will add a discussion on this issue.

Action: added in Discussion section: 'In the practical modeling, different periods of the same node may still be not independent. Therefore, the split scheme of such periods may be critical. For example, a period between two precipitation events can be treated as one sample, which can enhance independency between periods. Subsequently, such period can be divided into smaller time periods such as t, t-1, t-2, etc. to aggregate the node values to appropriate time scales. Thus one sample can represent the interaction relationship between variables with lags in this time period. Finally, we can integrate records of such periods between two precipitation events from sites across different climate zones and biomes to build synthesis models for global analysis of such problems. If further combined with the findings of process-based models, our understanding of climate and ecosystem interactions and feedback and their mechanisms in time is hopefully deepened.

'

L 308: Although the conclusions are free to mention related issues which are not part of the study, I would recommend to replace this last point e.g. by the importance of your findings for the modeling community.

Response: Thank you for your insightful comments. It will be replaced.

Action: Conclusion section is revised:

[revised manuscript text omitted]

---

## Referee Report (RR1)

**Review: Revisiting and attributing the global controls on terrestrial ecosystem functions of climate and plant traits at FLUXNET sites via causal graphical models**

**1. General comments**

The authors attempt to emulate hypothesized causal links between plant traits, climate and ecosystem functions by constructing a Bayesian Network (BN). Here links between traits and functions are based on expert knowledge, and the causality reflected in the BN is as good as the expert knowledge of the causal links. The authors then reevaluate the relative importance of plant traits and climate in determining ecosystem functions through a sensitivity analysis based mainly on FLUXNET data. Building on this they argue that climate indirectly affects ecosystem functions via its control on plant traits. We agree that, from an ecological perspective and considering the increasing availability of data, exploring different methods to analyze the interactions climate-vegetation involved in the ecosystem functions is a relevant and meaningful research topic.

The authors have carefully considered all the comments, questions and suggestions given during the first review. The issues of reproducibility and the lack of an appropriate model validation have been overcome for the most part by the authors with a k-fold cross-validation (Table S1), a better description of the data used (Table 1), appropriate references to justify the assignment of links to the BN (Table 2), and a clear specification of the discretization method and the reasoning behind it. In the supplementary material, the authors provided the validate confusion matrix; it can effectively confirm the effectiveness of the BN method in this study. The limitations of the study are objectively stated in the discussion section.

We are glad to see that most of our main concerns have been addressed, and we consider that the manuscript can be published after taking into account the few additional comments below we still have based on the revised manuscript.

**2. Specific Comments**

- The authors should not overstate the power of BNs to quantify causality. A little bit of caution is warranted here. We suggest that the authors include a statement in the abstract in which they recognize that the causality reflected in the BN is as good as the expert knowledge of the causal links.

- The criteria to assign the links involving climate variables were clarified significantly with Table 2 and appropriate references. Some references for the CSWI links are still missing: P -> CSWI and VPD -> CSWI. Based on the manuscript references, the reference Nelson et al. 2018 can support the assignment of these links. Please add this reference to Table 2.

- In Figure 5, the sentence "red for high values and in blue for low values" may cause some confusion if no further explanation is provided. Is the relationship between color depth and numerical values based solely on values or sorting?, and is the scale the same across the entire table?.

- In row 335, that is quite a meaningful way of predicting the future ecological properties at a global scale. However, it is quite difficult to achieve it due to the calculation cost. Many novel models based on BN and other causality networks could create more possibilities to deal with such complex real life situations (DOI: 10.1038/s42256-020-0218-x & DOI: 10.1038/s42256-022-00445-z). Maybe the authors would like to mention the state of the art here?

- The authors have provided a validation of their results through a k-fold cross-validation which will allow comparisons with future studies. However, it would be useful to clarify to the reader what is the criteria behind the selection of the variables ETmax, GPPsat, and NEPmax for the model validation. It should be made very clear early on that these are the primary objectives (i.e. the terminal nodes) of the study.

- Please verify that each table and figure is referred to in the text of the manuscript. Additionally, the manuscript still needs some writing and grammar improvements. An English check is required.

---

## Author Response (AR2)

**Reviewer #1**

The authors attempt to emulate hypothesized causal links between plant traits, climate and ecosystem functions by constructing a Bayesian Network (BN). Here links between traits and functions are based on expert knowledge, and the causality reflected in the BN is as good as the expert knowledge of the causal links. The authors then reevaluate the relative importance of plant traits and climate in determining ecosystem functions through a sensitivity analysis based mainly on FLUXNET data. Building on this they argue that climate indirectly affects ecosystem functions via its control on plant traits. We agree that, from an ecological perspective and considering the increasing availability of data, exploring different methods to analyze the interactions climate-vegetation involved in the ecosystem functions is a relevant and meaningful research topic.

The authors have carefully considered all the comments, questions and suggestions given during the first review. The issues of reproducibility and the lack of an appropriate model validation have been overcome for the most part by the authors with a k-fold cross-validation (Table S1), a better description of the data used (Table 1), appropriate references to justify the assignment of links to the BN (Table 2), and a clear specification of the discretization method and the reasoning behind it. In the supplementary material, the authors provided the validate confusion matrix; it can effectively confirm the effectiveness of the BN method in this study. The limitations of the study are objectively stated in the discussion section.

We are glad to see that most of our main concerns have been addressed, and we consider that the manuscript can be published after taking into account the few additional comments below we still have based on the revised manuscript.

Response: Thank you for your insightful comments, which have been very helpful in improving the manuscript. We revised this manuscript in accordance with your few additional comments.

**Specific Comments**

- The authors should not overstate the power of BNs to quantify causality. A little bit of caution is warranted here. We suggest that the authors include a statement in the abstract in which they recognize that the causality reflected in the BN is as good as the expert knowledge of the causal links.
Response: added in Abstract: 'The causality reflected in the BN is as good as the expert knowledge of the causal links.'

- The criteria to assign the links involving climate variables were clarified significantly with Table 2 and appropriate references. Some references for the CSWI links are still missing: P -> CSWI and VPD -> CSWI. Based on the manuscript references, the reference Nelson et al. 2018 can support the assignment of these links. Please add this reference to Table

2.
Response: P -> CSWI and VPD -> CSWI links and the reference Nelson et al. 2018 were added in Table 2.

- In Figure 5, the sentence "red for high values and in blue for low values" may cause some confusion if no further explanation is provided. Is the relationship between color depth and numerical values based solely on values or sorting?, and is the scale the same across the entire table?.
Response: added in the figure caption: 'Color depth is dependent on values and the scale is the same in each row'.

- In row 335, that is quite a meaningful way of predicting the future ecological properties at a global scale. However, it is quite difficult to achieve it due to the calculation cost. Many novel models based on BN and other causality networks could create more possibilities to deal with such complex real life situations (DOI: 10.1038/s42256-020-0218-x & DOI: 10.1038/s42256-022-00445-z). Maybe the authors would like to mention the state of the art here?
Response: elaborated as 'Such research frameworks in BN-based modeling may be difficult due to high computational costs given the large amount of data. Fortunately, recently proposed new causal models have the potential to address this limitation, such as the introduction of causality into deep learning frameworks (Luo et al., 2020; Cui and Athey, 2022).'

- The authors have provided a validation of their results through a k-fold cross-validation which will allow comparisons with future studies. However, it would be useful to clarify to the reader what is the criteria behind the selection of the variables ETmax, GPPsat, and NEPmax for the model validation. It should be made very clear early on that these are the primary objectives (i.e. the terminal nodes) of the study.
Response: elaborated in methods: 'These three nodes are the main terminal nodes and primary objectives of the BN and represent the main water and carbon-related ecosystem functions, respectively. The accuracy of these three variables can largely reflect the overall performance of BN.'

- Please verify that each table and figure is referred to in the text of the manuscript. Additionally, the manuscript still needs some writing and grammar improvements. An English check is required.
Response: Table and figure were verified. The Table 2 was not referred to and we added 'The explanation of added causal links was shown in Table 2.' Writing and grammar issues were also improved.

**Reviewer #2**

The revision includes a major effort by clarifying methods, approach and the representation and interpretation of the results. Especially, the building of the statistical model and the inclusion of previous knowledge became clearer and comprehensible. The text benefits from the removal of redundant paragraphs and sharpening of the conclusions. With these revisions only minor text corrections are necessary before publication.

Response: Thank you for your insightful comments, which have been very helpful in improving the manuscript. We revised this manuscript in accordance with your few additional comments.

Minor remarks:
There are still some remaining misspellings and text fragments so that an independent reader could go through the text. Some examples of needed edits are listed here:
L 272: Please explain what you mean with ' is relatively reliable' for the BN-Based analysis.
Revised: 'relatively reliable in terms of the represented mechanisms of causal links'

L. 286: You refer repeatedly to the reference study Migliavacca et al. (2021) but please do not use abbreviated words like here the 'ref.' and replace these references by the citation.
Revised: 'in the ref. (Migliavacca et al., 2021)' was replaced with 'in Migliavacca et al., 2021'

L. 287: Please exchange twice in this line 'chosed' with 'chosen'.
Replaced.